# Nodal and planar cell polarity signaling cooperate to regulate zebrafish convergence and extension gastrulation movements

**Margot LK Williams†\*, Lilianna Solnica-Krezel**

Department of Developmental Biology, Washington University School of Medicine, St. Louis, United States

**Abstract** During vertebrate gastrulation, convergence and extension (C and E) of the primary anteroposterior (AP) embryonic axis is driven by polarized mediolateral (ML) cell intercalations and is influenced by AP axial patterning. Nodal signaling is essential for patterning of the AP axis while planar cell polarity (PCP) signaling polarizes cells with respect to this axis, but how these two signaling systems interact during C and E is unclear. We find that the neuroectoderm of Nodal-deficient zebrafish gastrulae exhibits reduced C and E cell behaviors, which require Nodal signaling in both cell- and non-autonomous fashions. PCP signaling is partially active in Nodal-deficient embryos and its inhibition exacerbates their C and E defects. Within otherwise naïve zebrafish blastoderm explants, however, Nodal induces C and E in a largely PCP-dependent manner, arguing that Nodal acts both upstream of and in parallel with PCP during gastrulation to regulate embryonic axis extension cooperatively.

**\*For correspondence:**
Margot.Williams@BCM.edu

**Present address:** †Center for Precision Environmental Health, Department of Molecular and Cellular Biology, Baylor College of Medicine, Houston, United State

**Competing interests:** The authors declare that no competing interests exist.

## Introduction

The embryonic body plan first emerges during gastrulation, when the three primordial germ layers — ectoderm, mesoderm, and endoderm — are formed and shaped, and embryonic axes are morphologically manifest. At this time, the anteroposterior (AP) body axis undergoes dramatic extension, a process that is essential for proper body plan formation and neural tube closure (*Wallingford and Harland, 2002*; *Davidson and Keller, 1999*). Axial extension results from highly conserved convergence and extension (C and E) movements that simultaneously elongate tissues along the AP axis and narrow them in the orthogonal mediolateral (ML) dimension (*Keller et al., 2000*; *Warga and Kimmel, 1990*; *Keller and Danilchik, 1988*). This process is driven in vertebrate embryos by a combination of highly polarized cell behaviors, including mediolateral intercalation behavior (MIB) and directed migration (*Keller et al., 2000*; *Warga and Kimmel, 1990*). MIB entails the ML alignment and elongation of cells and the acquisition of bipolar protrusive behavior through which cells intercalate in a polarized fashion between their anterior and posterior neighbors (*Shih and Keller, 1992a*; *Shih and Keller, 1992b*).

This ML polarization of cells and their behaviors requires planar cell polarity (PCP) signaling (*Wallingford et al., 2000*; *Park and Moon, 2002*; *Jessen et al., 2002*; *Wang et al., 2006a*; *Wang, 2006b*; *Ybot-Gonzalez et al., 2007*; *Heisenberg et al., 2000*; *Kilian et al., 2003*; *Topczewski et al., 2001*; *Tada and Smith, 2000*). First discovered in *Drosophila*, this conserved signaling network is essential for collective polarity across cellular fields, within the plane of a tissue (*Vinson and Adler, 1987*; *Strutt and Strutt, 2005*). Core PCP components acquire asymmetric distribution within cells (*Bastock et al., 2003*), with some becoming enriched at the anterior or posterior aspects of vertebrate cells as they undergo gastrulation movements (*Wang, 2006b*;

*Roszko et al., 2015*; *Ciruna et al., 2006*; *Yin et al., 2008*). Because impairment of PCP signaling dramatically disrupts the polarized cell behaviors underlying axis extension but has little effect on patterning, it is thought to act as a molecular compass that allows cells to sense and/or respond to positional cues within the embryo (*Yin et al., 2009*; *Gray et al., 2011*). This implies the existence of a molecular mechanism by which patterning information is communicated to this compass, and ultimately to the cellular machinery that drives polarized C and E cell behaviors.

In contrast with vertebrate embryos, PCP signaling is not essential for axial extension in *Drosophila*, which instead requires AP patterning that confers the striped expression of pair-rule genes (*Irvine and Wieschaus, 1994*; *Zallen and Wieschaus, 2004*). These in turn regulate the expression of Toll-like receptors in a partially overlapping striped pattern, comprising a positional code along the extending AP axis (*Paré et al., 2014*). AP patterning is similarly a prerequisite for extension of the gut tube in *Drosophila* and *Xenopus*, and during *Xenopus* gastrulation (*Ninomiya et al., 2004*; *Johansen et al., 2003*; *Li et al., 2008*). In particular, *Ninomiya et al., 2004* reported that *Xenopus* gastrula explants with different AP positional values extend when apposed ex vivo, whereas those with the same positional identity do not. Notably, these positional values could be recapitulated in explants by different doses of the TGFβ ligand Activin (*Ninomiya et al., 2004*), which signals largely via the Nodal signaling pathway during early vertebrate embryogenesis (*Pauklin and Vallier, 2015*). These results demonstrate that AP patterning is required for axial extension ex vivo and implies a crucial role for Nodal signaling at this intersection of tissue patterning and morphogenesis in vivo.

Nodal is a TGFβ-superfamily morphogen whose graded signaling within the embryo produces discrete developmental outcomes depending on a cell's position within that gradient and the resulting signaling level/duration to which it is exposed (*Dyson and Gurdon, 1998*; *Gurdon et al., 1999*; *van Boxtel et al., 2015*; *Dubrulle et al., 2015*; *Chen and Schier, 2001*). Upon binding of Nodal–Gdf3 (Vg1) heterodimers (*Pelliccia et al., 2017*; *Bisgrove et al., 2017*; *Montague and Schier, 2017*), the receptor complex — comprised of two each of the Type I and Type II serine-threonine kinase receptors Acvr1b and Acvr2b and the co-receptor Tdgf — is activated and phosphorylates the downstream transcriptional effectors Smad2 and/or Smad3 (*Gritsman et al., 1999*; *Schier and Shen, 2000*). Nodal signaling is essential for specification of endoderm and mesoderm germ layers and their patterning along the AP axis, with the highest signaling levels producing endoderm and the most dorsal/anterior mesoderm fates (*Thisse et al., 2000*; *Gritsman et al., 2000*; *Vincent et al., 2003*; *Dougan et al., 2003*; *Feldman et al., 1998*; *Feldman et al., 2000*). Mouse embryos that are mutant for Nodal signaling components fail to gastrulate, resulting in early embryonic lethality (*Conlon et al., 1994*). Nodal-deficient zebrafish undergo highly abnormal gastrulation, failing to specify endoderm and most mesoderm (*Dubrulle et al., 2015*; *Gritsman et al., 1999*; *Feldman et al., 1998*), resulting in embryos that are comprised largely of neuroectoderm and displaying severe neural tube and axis extension defects (*Aquilina-Beck et al., 2007*; *Gonsar et al., 2016*).

Restoration of mesoderm to maternal-zygotic *one-eyed pinhead* (MZ*oep*) zebrafish mutants, which lack the essential Tdgf Nodal co-receptor (*Gritsman et al., 1999*), improves AP axis length and the morphology of the neural tube (*Araya et al., 2014*), implying that Nodal promotes C and E of the neuroectoderm non-autonomously via specification of mesoderm. However, additional evidence points to a more direct role for Nodal signaling in C and E cell behaviors. First, Activin signaling via Nodal receptors is sufficient for C and E of *Xenopus* animal cap explants (*Ninomiya et al., 2004*; *Symes and Smith, 1987*; *Howard and Smith, 1993*) and for the underlying planar polarity of cells (*Shindo et al., 2008*). Furthermore, knockdown of two out of six *Xenopus* Nodal ligands disrupts C and E movements without affecting mesoderm specification (*Luxardi et al., 2010*). Nodal and Activin were also shown to promote translocation of the core PCP component Disheveled to cell membranes, suggesting that it acts upstream of PCP signaling activation (*Ninomiya et al., 2004*; *Trichas et al., 2011*). Further evidence suggests that AP patterning is required in addition to PCP for C and E morphogenesis (*Ninomiya et al., 2004*), and while such patterning can be recapitulated by graded exposure of explants to Activin, it is not known whether Nodal and/or other signals play this role in vivo. Therefore, how Nodal interfaces with the PCP molecular compass during gastrulation remains to be determined.

Here, we investigate the role of Nodal signaling in C and E gastrulation movements in zebrafish. We demonstrate that defective C and E movements in the neuroectoderm of MZ*oep* mutant gastrulae are associated with reduced ML cell alignment and protrusive activity. Transplantation of mutant

cells into the prospective neuroectoderm of wild-type (WT) embryos only partially restored their ML polarity during gastrulation, demonstrating both cell-autonomous and non-autonomous roles for Nodal in planar cell polarization. Surprisingly, MZ*oep*$^{-/-}$ neuroectoderm cells exhibited normal, anteriorly biased localization of Prickle-GFP, a hallmark of PCP signaling activity. Consistent with active PCP signaling in the absence of Nodal, C and E defects in MZ*oep* mutants were exacerbated by interference with the core PCP component Vangl2. To examine further this cell-autonomous function of Nodal signaling in morphogenesis, we employed zebrafish blastoderm explantation to isolate the effects of Nodal from endogenous signaling centers of intact embryos. We found that, as for Nodal and Activin in *Xenopus* animal cap assays, expression of Nodal ligands was sufficient to induce robust, PCP-dependent ML cell polarization and C and E of naïve zebrafish blastoderm explants in culture. Treatment of explants with a Nodal inhibitor revealed a continuous requirement for Nodal signaling in ex vivo extension after mesoderm was specified and even in the absence of mesoderm, implying a primary, mesoderm-independent role for Nodal in C and E. Together, these data support a model in which Nodal signaling promotes ML cell polarity and C and E, both upstream and independent of PCP signaling, and predicts additional AP patterning mechanisms that instruct the PCP compass during vertebrate gastrulation.

## Results

### Nodal regulates C and E cell behaviors cell-autonomously and non-autonomously

Zebrafish embryos that are double mutant for the two *nodal*-related genes expressed during gastrulation, *ndr1* (*sqt*) and *ndr2* (*cyc*), or that lack both maternal and zygotic function of the co-receptor Tdgf (MZ*oep*$^{-/-}$) or the downstream effector Smad2, exhibit severe dorsolateral mesendoderm deficiencies and impaired AP extension of the enlarged neuroectoderm (*Dubrulle et al., 2015*; *Gritsman et al., 1999*; *Feldman et al., 1998*; *Figure 1A*). However, underlying cell behavior defects during gastrulation have not been fully characterized. We therefore analyzed cell movements in the dorsal region of WT and MZ*oep* mutants by time-lapse confocal microscopy for a period of three hours, beginning shortly after the onset of C and E movements (80% epiboly, 8.5 hr post-fertilization [hpf]). Automated tracking of fluorescently labeled nuclei in WT gastrulae revealed clear convergence of cells from lateral positions toward the dorsal midline, and concomitant extension along the AP axis (*Figure 1B–C*, top, *Video 1*). Analysis of cell velocities demonstrated that rates of cell movement were highest in the lateral-, anterior-, and posterior-most regions of the gastrula and lowest in the center (*Figure 1D*, top). This is consistent with mediolateral intercalation, which is characterized by a stationary point near the embryo's equator and cell velocities that increase proportionally with their distance from this point (*Glickman et al., 2003*; *Concha and Adams, 1998*). MZ*oep* mutant gastrulae, by contrast, exhibited disorganized cell movement and velocity patterns that are inconsistent with ML intercalation (*Figure 1B–D*, bottom). These cells moved along swirling paths, which contrasted with the direct anterior-, posterior-, and medial-ward movement of WT cells, and were seen to cross the dorsal midline, which was not observed in WT embryos (*Figure 1B*; *Concha and Adams, 1998*).

Both C and E movements were apparent in WT gastrulae when cell track displacement in the ML or AP dimension was plotted against the starting position of each cell along these respective axes (*Figure 1E–F*). For example, cells on the left side of each embryo exhibited right-ward displacement and vice versa as they converged toward the midline, resulting in a negative slope (*Figure 1E*, blue). Meanwhile, cells in the anterior and posterior of each embryo moved anteriorly and posteriorly, respectively, extending the AP axis and yielding a positive slope (*Figure 1F*, blue). Cell track displacement in MZ*oep*$^{-/-}$ gastrulae, on the other hand, was not neatly graded along the ML and AP embryonic axes as observed in WT embryos (*Figure 1E–F*, purple). Although convergence was still apparent in these mutants (*Figure 1E*), displacement of MZ*oep*$^{-/-}$ cells in the ML dimension was significantly reduced compared to WT (*Figure 1G*, p<0.0001, Kolmogorov-Smirnov [K-S] test), and AP extension was particularly severely disrupted in terms of both absolute displacement and spatial organization of cell movements (*Figure 1F–G*). Although these results are consistent with previous findings that convergence movements are observed within Nodal-deficient gastrulae despite an almost complete lack of extension (*Gritsman et al., 1999*; *Feldman et al., 1998*), they demonstrate

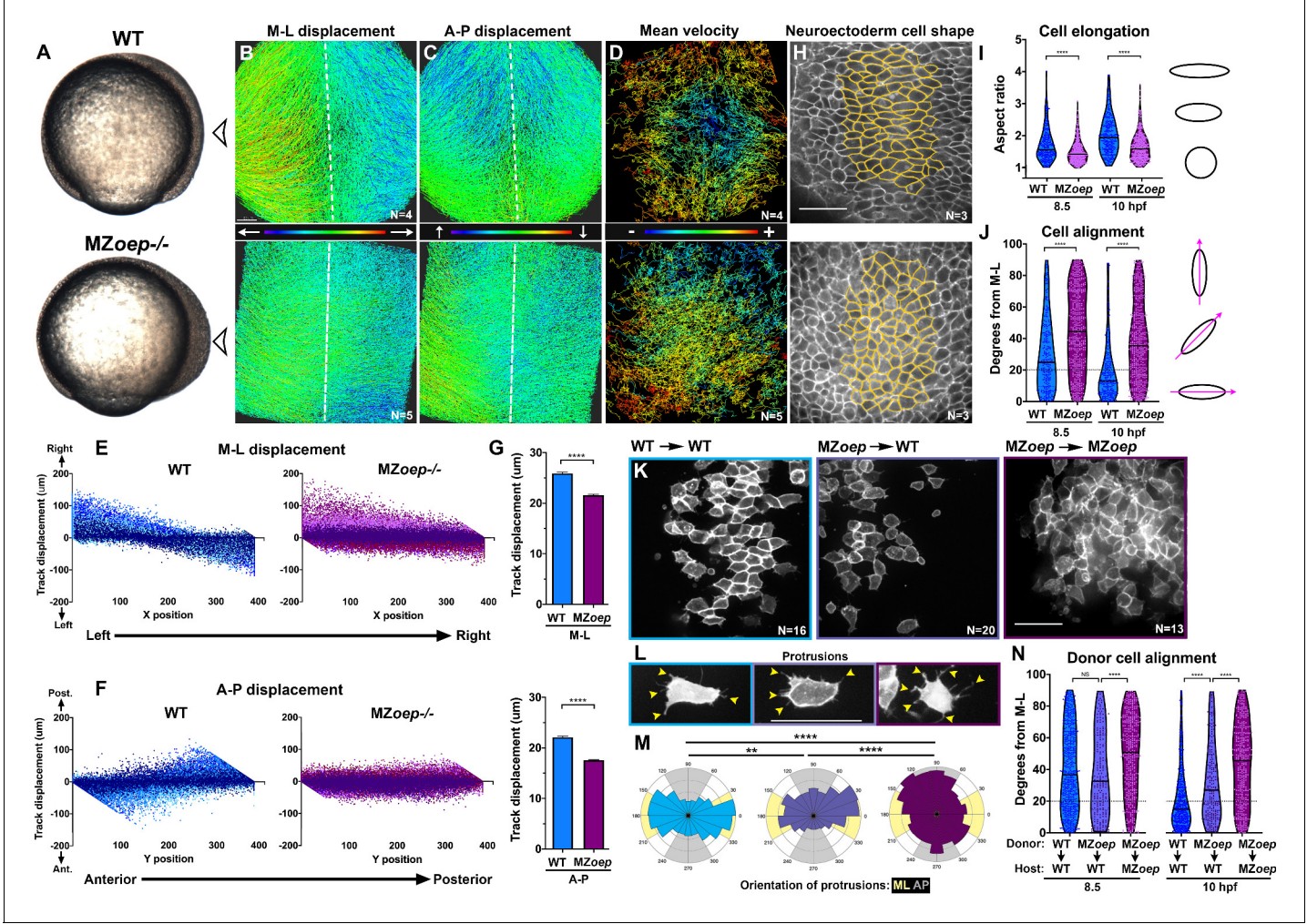

**Figure 1.** Nodal signaling regulates convergence and extension gastrulation cell behaviors. (**A**) Bright-field images of live WT and MZ*oep*⁻/⁻ embryos at 80% epiboly (8.5 hpf). The arrowheads indicate the point of view (dorsal side) for all fluorescent confocal micrographs. (**B–D**) Representative images of automated tracking of fluorescently labeled nuclei in the dorsal hemisphere of WT (top) and MZ*oep*⁻/⁻ (bottom) gastrulae. Tracks represent cell movements over three hours of time-lapse confocal imaging, beginning at 8.5 hpf, and are colored according to their displacement in the mediolateral (**B**) and anteroposterior (**C**) dimensions or the mean velocity of cell movement (**D**). Dotted lines indicate dorsal midline. (**E, F**) Displacement of cell tracks in the mediolateral (**E**) and anteroposterior (**F**) dimensions in WT (blue) and MZ*oep*⁻/⁻ (purple) gastrulae [as shown in (**B–D**)]. Each dot represents a single cell track, each color represents an individual embryo, N = 4 WT and 5 MZ*oep*⁻/⁻. (**G**) Absolute displacement of cell tracks in ML (top) and AP (bottom) dimensions. Bars are mean with 95% confidence interval, p<0.0001, Kolmogorov-Smirnoff (K-S) tests. (**H**) Representative images of membrane-labeled neuroectoderm in live WT (top) and MZ*oep*⁻/⁻ (bottom) gastrulae with cells outlined in yellow. (**I, J**) Neuroectoderm cell elongation (**I**) and alignment (**J**) at 8.5 hpf (left) and 10 hpf (right). Each dot represents a single cell, black bars are mean values in (**I**), and median values in (**J**). N = 3 embryos of each genotype, p<0.0001, Mann-Whitney test in (**I**), K-S test in (**J**). (**K**) Representative images of membrane-labeled donor cells of the indicated genotypes within the neuroectoderm of unlabeled host gastrulae. N indicates the number of embryos analyzed from three independent trials. (**L**) Representative images of protrusions (arrowheads) made by transplanted neuroectoderm cells of the genotypes/conditions indicated in (**K**). (**M**) The orientation of all protrusions between 8.5 and 10 hpf is shown in radial histograms divided into 20° bins, with 0 and 180 representing the ML axis. Yellow and gray quadrants represent ML- and AP-oriented protrusions, respectively. **, p=0.0053; ****, p<0.0001; Chi-square. (**N**) Alignment of donor cells as in (**J**). The number of embryos in each condition is indicated in the corresponding panels in (**K**). Anterior is up in all images, scale bars are 50 μm. Dotted lines in (**J, N**) show 20 degrees from ML for reference.

that convergence movements are also reduced and disorganized in MZ*oep* mutant gastrulae (*Figure 1E–G*).

We next used a fluorescent membrane marker to assess the ML cell elongation, alignment and protrusive activity underlying C and E in the neuroectoderm of WT and MZ*oep*⁻/⁻ embryos from 8.5 hpf until the end of gastrulation (10 hpf) (*Figure 1H–M*). Cell elongation is represented as the aspect

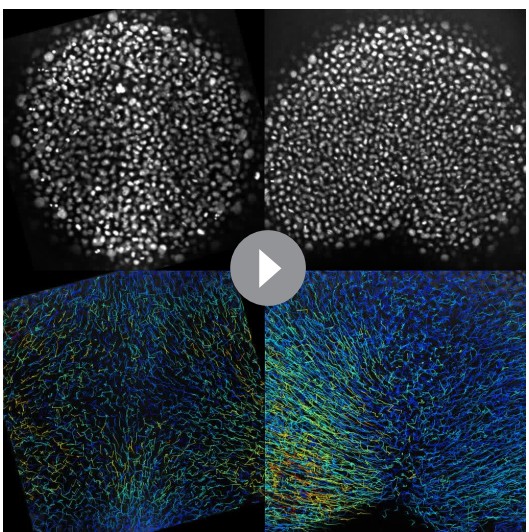

**Video 1.** Automated tracking of fluorescent nuclei in live zebrafish gastrulae. Time-lapse confocal series from approximately 8.5 to 11.5 hpf in representative WT (left) and MZ*oep*⁻/⁻ (right) gastrulae injected with *H2B-RFP* RNA. Cell tracks shown below are colored according to track displacement, with warmer colors indicating higher displacement.

https://elifesciences.org/articles/54445#video1

ratio of each cell (major/minor cell axis), and cell alignment was measured as the orientation of each major cell axis with respect to the embryo's ML axis, with 0° indicating perfect ML orientation. We found that WT neuroectoderm cells were significantly more elongated than MZ*oep*⁻/⁻ cells throughout gastrulation (*Figure 1I*; p<0.0001, Mann-Whitney tests). Unlike the marked ML alignment of WT cells that increased over time (*Figure 1J*, blue), MZ*oep*⁻/⁻ cells were significantly less well aligned at both 8.5 and 10 hpf time points (*Figure 1J*, purple) (p<0.0001, K-S test). We then measured the orientation of cellular protrusions within the neuroectoderm of WT and MZ*oep*⁻/⁻ gastrulae using cell transplantation to achieve sparse labeling (*Figure 1K–M*). Protrusions made by WT cells exhibited a strong ML bias typical of MIB (*Figure 1L–M*; *Keller et al., 2000*), whereas protrusions of MZ*oep*⁻/⁻ cells were essentially randomly oriented with a slight anterior bias (*Figure 1L–M*). Together, these results demonstrate a severe disruption of C and E movements and polarized cell behaviors in MZ*oep* mutant gastrulae.

Previous studies have shown that axis extension can be largely rescued by restoration of mesoderm to MZ*oep* mutant embryos (*Araya et al., 2014*), but the autonomy of Nodal signaling within the neuroectoderm has not been examined at the level of cell polarity. To determine whether Nodal regulates ML alignment cell-autonomously within the neuroectoderm, we transplanted membrane-labeled MZ*oep*⁻/⁻ cells into the prospective neuroectoderm of WT host embryos (*Figure 1K*). We then measured donor cell alignment at late gastrulation (10 hpf) and found that MZ*oep*⁻/⁻ cells within WT hosts were significantly less well aligned than WT control donors (*Figure 1N*; p<0.0001, K-S test). At mid-gastrulation (8.5 hpf), some experiments showed that the alignment of MZ*oep*⁻/⁻ donor cells was not significantly different from that of WT donor cells in WT hosts (*Figure 1N*), whereas other experiments indicated that MZ*oep*⁻/⁻ donor cells were significantly less well aligned than WT controls at this stage (*Figure 2D*; p<0.001, K-S test). Because cell alignment in unmanipulated WT gastrulae (*Figure 1J*) more closely resembles that of the WT control donors shown in *Figure 2D*, we conclude that the alignment of MZ*oep*⁻/⁻ cells within WT hosts is reduced compared to WT control donors. Notably, mutant cells in WT host gastrulae were significantly better aligned than MZ*oep*⁻/⁻ cells in MZ*oep*⁻/⁻ hosts at both time points (*Figure 1N*) (p<0.0001, K-S test). Orientation of MZ*oep*⁻/⁻ cellular protrusions was also partially improved within WT hosts (*Figure 1L–M*), as the distribution of these protrusions differed significantly from that in MZ*oep*⁻/⁻ controls but did not align to the same degree as in WT cells (p<0.0001 and p=0.0053, respectively, Chi-square test). Together these results reveal an essential role for Nodal signaling in neuroectoderm C and E cell behaviors, including both non-autonomous and cell-autonomous functions during ML cell polarization.

## Nodal functions partially in parallel with PCP signaling during axis extension

The reduced ML polarity of MZ*oep*⁻/⁻neuroectoderm cells resembles PCP mutant phenotypes (*Jessen et al., 2002*; *Kilian et al., 2003*; *Topczewski et al., 2001*; *Ulrich et al., 2003*), raising the possibility that loss of PCP signaling may underlie C and E defects in MZ*oep* mutants. We compared gene expression in WT and MZ*oep*⁻/⁻ gastrulae at 90% epiboly (~9 hpf) by RNA-sequencing and found that of the genes with known roles in PCP signaling in zebrafish, only one (*prickle1b*) exhibited altered expression in MZ*oep* mutants (*Figure 2—figure supplement 1*). Accordingly, *wnt5b, vangl2*

(*trilobite*), and *gpc4* (*knypek*) transcripts were all detected in MZ*oep*$^{-/-}$ gastrulae by whole mount in situ hybridization (WISH) (*Figure 2—figure supplement 1*). Although this suggests that Nodal signaling does not regulate PCP at the level of gene expression, we hypothesized that it could instead regulate the activity of these signaling components. We therefore assessed PCP signaling activity by examining the intracellular localization of the core PCP component Prickle fused to GFP (Pk-GFP), whose association with anterior cell membranes is indicative of PCP activation and polarization (*Ciruna et al., 2006*; *Yin et al., 2008*), using transplantation to achieve sparse labeling of neuroectoderm cells. We found that WT cells (in WT hosts) and MZ*oep*$^{-/-}$ cells (in MZ*oep*$^{-/-}$ hosts) exhibited similar proportions of anteriorly localized Pk-GFP puncta, although MZ*oep* mutant cells contained significantly more membrane-associated puncta that were not anteriorly localized (chi-square, p=0.0001) (*Figure 2A, B*), suggesting that PCP signaling is largely active in Nodal signaling-deficient gastrulae.

Because PCP signaling establishes planar polarity via intra- and inter-cellular interactions between its molecular components (*Goodrich and Strutt, 2011*; *Bayly and Axelrod, 2011*), we hypothesized that the partial ML polarization of MZ*oep*$^{-/-}$ cells observed upon transplantation into WT hosts indicates the ability of Nodal-deficient cells to respond to host PCP signaling (*Figure 1L, N*). To test this, we disrupted PCP signaling in MZ*oep*$^{-/-}$ embryos using an antisense morpholino oligonucleotide (MO) against *vangl2* (*Williams et al., 2012*) that phenocopies C and E defects of *trilobite/vangl2* mutants (*Figure 2—figure supplement 1*; *Solnica-Krezel et al., 1996*), and transplanted cells from these MZ*oep*$^{-/-}$;*vangl2* morphant donors into the prospective neuroectoderm of WT hosts. At both

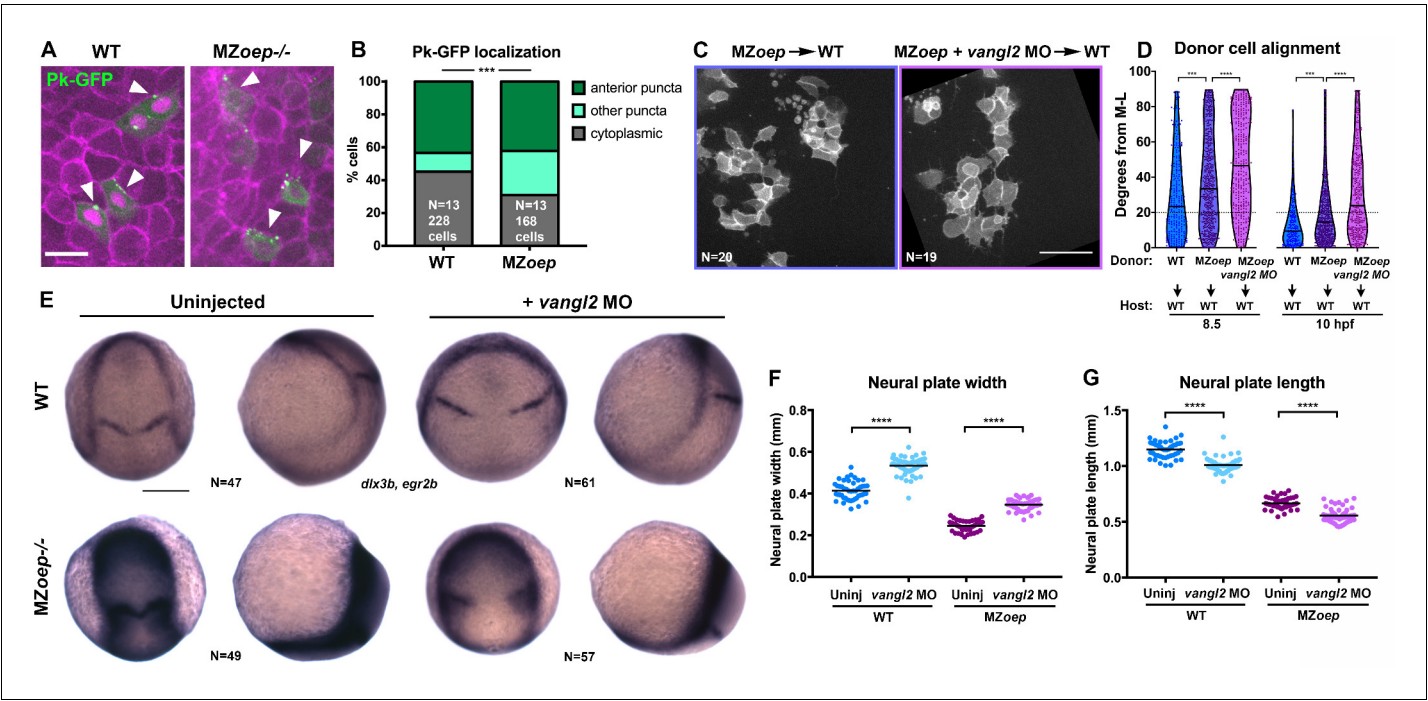

**Figure 2.** PCP signaling is active and contributes to C and E in Nodal signaling mutants. (**A**) Representative images of transplanted Prickle (Pk)-GFP donor cells (co-expressing H2B-RFP) within the neuroectoderm of membrane-labeled WT and MZ*oep*$^{-/-}$ host gastrulae. Arrowheads indicate puncta at anterior edges. (**B**) Pk-GFP localization in the genotypes indicated. N indicates the number of embryos and cells analyzed for each condition from four independent trials, p<0.001, Chi-square test. (**C**) Representative images of membrane-labeled MZ*oep*$^{-/-}$ donor cells without (left) and with (right) 2 ng MO4-*vangl2* transplanted into the neuroectoderm of unlabeled host gastrulae from five independent trials. (**D**) Donor cell alignment as in *Figure 1*. The number of embryos in each condition is indicated on the corresponding panels in (**C**), WT→WT control N = 10. ***, p<0.001; ****, p<0.0001; K-S tests. (**E**) Whole mount in situ hybridization (WISH) for *dlx3b* and *egr2b* in WT (top) and MZ*oep*$^{-/-}$ (bottom) gastrulae at 9.5 hpf, uninjected or injected with 2 ng MO4-*vangl2*. Dorsal views on the left, lateral views on the right. (**F, G**) Width (**F**) and length (**G**) of neural plates in the embryos depicted in (**E**). Each dot represents a single embryo, black bars are mean values. Number of embryos in each condition is indicated on the corresponding panel in (**E**), p<0.0001, Unpaired T-tests. Anterior is up in all images, scale bar is 20 µm in (**A**), 50 µm in (**C**), and 200 µm in (**E**).

The online version of this article includes the following figure supplement(s) for figure 2:

**Figure supplement 1.** PCP signaling in Nodal-deficient embryos.

mid- and late gastrulation stages, ML alignment was significantly reduced in MZ*oep*^−/−;*vangl2* morphant cells compared with control MZ*oep*^−/−cells (*Figure 2C, D*), further supporting the notion that PCP is active in the absence of Nodal signaling. Notably, in this series of experiments, we found that MZ*oep*^−/− donor cells were significantly less well aligned than WT control donors at both 8.5 and 10 hpf (*Figure 2D*). Together with the mosaic analyses described above and in *Figure 1K–N*, these results reveal a cell-autonomous role for Nodal in ML cell polarization throughout gastrulation. Finally, we tested whether disrupting PCP signaling using a *vangl2* MO reduced axis extension in MZ*oep* mutant gastrulae. We found that the neural plates of both WT and MZ*oep* mutants, as marked by expression of *dlx3b,* were significantly wider and shorter upon injection with *vangl2* MO than in uninjected controls at late gastrulation stages (*Figure 2E–G*), indicating reduced C and E. These results provide further evidence that PCP signaling is active and contributes to C and E movements in embryos lacking Nodal signaling. We noted that WISH staining for *dlx3b* was noticeably darker in MZ*oep*^−/−gastrulae (*Figure 2E*), likely reflecting a slight increase in expression levels (*Figure 2—figure supplement 1*) compounded by increased cell density resulting from reduced extension of the neuroectoderm in these mutants (*Figure 1*; *Gritsman et al., 1999*). Together, these results indicate that Nodal does not regulate C and E solely via PCP, and that PCP is not activated or polarized strictly downstream of Nodal.

## Nodal signaling promotes ex vivo extension and tissue patterning

We have demonstrated that Nodal signaling is necessary for full planar polarization of cells and cell behaviors underlying C and E gastrulation movements. To test whether Nodal is also sufficient for these behaviors, we sought to define the role of Nodal during axis extension in relative isolation, independent of other signaling and patterning events within the embryo. To this end, we employed blastoderm explantation, a technique that is used to excise only the animal-most region of the blastoderm from ~2.5 hpf zebrafish embryos, thereby isolating this region from endogenous signaling centers at the embryonic margin and producing clusters of relatively naïve cells that can be grown and manipulated in culture (*Figure 3A*; *Xu et al., 2014*; *Sagerström et al., 1996*; *Schauer et al., 2020*; *Trivedi et al., 2019*). To determine the effect of Nodal signaling on such explants, we injected single-celled WT embryos with synthetic *ndr2* mRNA at doses from 2.5 to 100 pg per embryo, explanted the animal half of each blastoderm at the 256–512 cell stage (2.5 hpf), and cultured these explants ex vivo until intact siblings reached early segmentation stages (*Figure 3A*, *Figure 3—figure supplement 1*). We found that several of these doses induced robust extension of explants in culture, whereas explants cut from *GFP*-injected or uninjected WT control or from *ndr2*-injected MZ*oep*^−/− embryos failed to extend (*Figure 3B–E*, *Figure 3—figure supplement 1*).

Time-lapse imaging of live *ndr2*-injected explants revealed the onset of extension morphogenesis at or around 8 hpf (*Figure 3F*, *Video 2*), corresponding with the start of C and E movements in intact embryos (*Sepich et al., 2005*). All *ndr2* RNA doses induced some degree of extension over control explants, but the intermediate doses (5–25 pg) were most effective, with 10 pg producing the most extension and the highest dose tested (100 pg) producing the least (*Figure 3—figure supplement 1*). Therefore, 10 pg *ndr2* was used for most subsequent experiments. Automated nuclear tracking within *ndr2*-expressing explants revealed patterns of cell movement that were characteristic of C and E by MIB (*Figure 3G–H*), with the axis of explant extension defined as 'AP' and the orthogonal axis as 'ML'. Indeed, plots of cell track displacement in the 'ML' and 'AP' dimensions (*Figure 3I–J*, blue) yielded the same negative and positive slopes, respectively, observed in intact embryos (*Figure 1*). These results demonstrate both convergence and extension movements within Nodal-expressing explants, although extension was more prominent (*Figure 3K*). Although cell movement was detected in *RFP*-injected control explants, cell track displacement was not spatially organized (*Figure 3I–J*, gray) and was significantly reduced compared to that in *ndr2*-expressing siblings (*Figure 3K*). We also examined cell divisions within *ndr2*-expressing explants to address the possibility that differential proliferation contributes to ex vivo extension. Although divisions were preferentially localized along the axis of extension, they were not concentrated in any particular region of the explant, and the number of cell divisions detected at these stages was relatively small (*Figure 3—figure supplement 2*). This suggests that, as in intact zebrafish gastrulae (*Liu et al., 2017*), cell proliferation is not likely to be a major driver of C and E morphogenesis ex vivo, consistent with a recent report that blocking cell divisions in zebrafish blastoderm explants did not prevent their extension (*Schauer et al., 2020*).

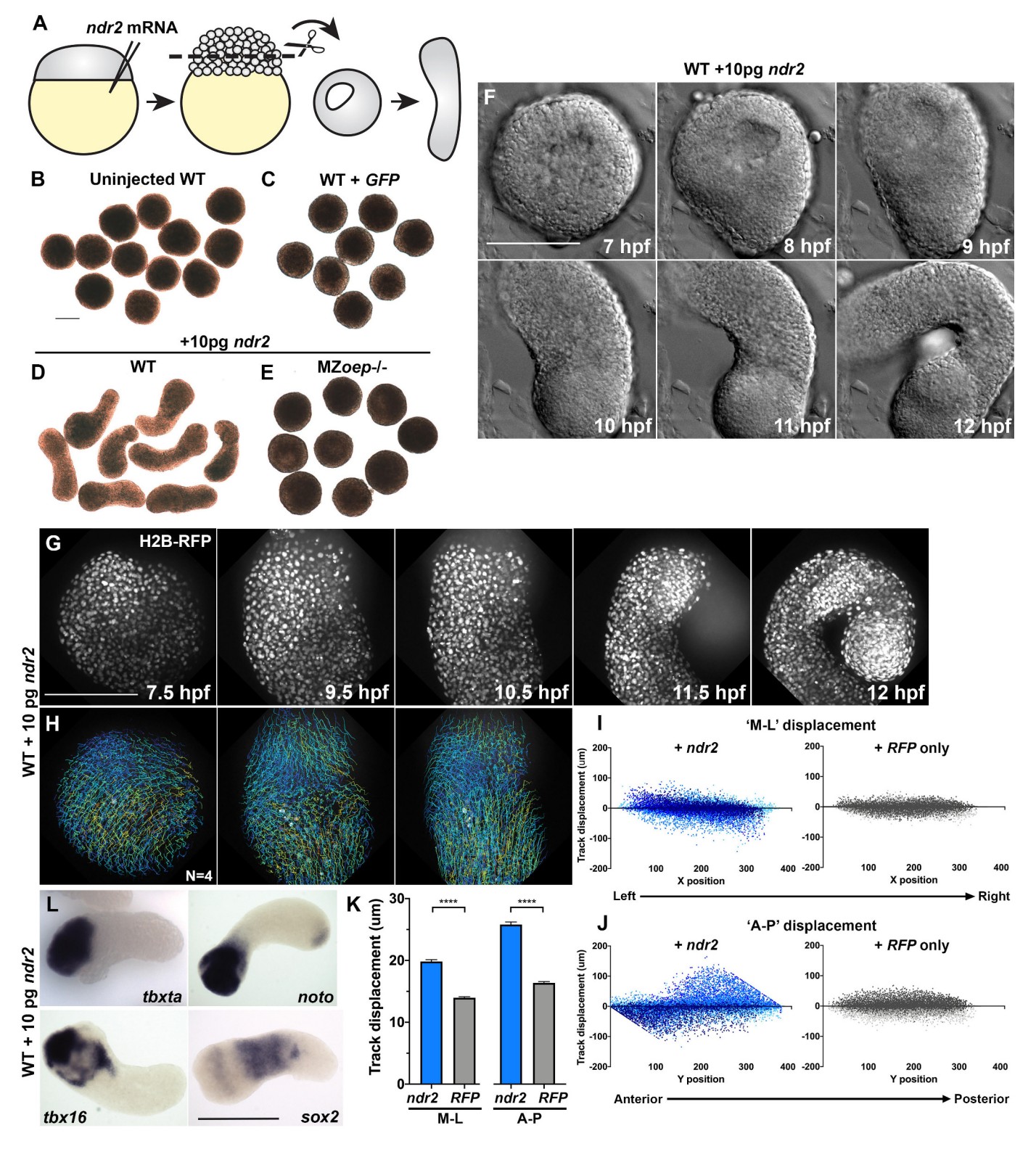

**Figure 3.** Nodal ligands promote ex vivo C and E of blastoderm explants. (**A**) Diagram of injection and explantation of zebrafish embryos. (**B–E**) Representative bright-field images of live blastoderm explants of the indicated conditions/genotypes at the equivalent of the 2–4 somite stage. (**F**) Time-lapse DIC series of a representative explant from a WT embryo injected with 10 pg *ndr2* RNA. (**G, H**) Time-lapse series of H2B-RFP labeled nuclei (**G**) and automated cell tracking (**H**) within a representative explant from a WT embryo injected with 10 pg *ndr2* RNA. Tracks represent cell movements
*Figure 3 continued on next page*

*Figure 3 continued*

over 3.5 hr of time-lapse confocal imaging beginning at 7.5 hpf and are colored according to mean track displacement. (**I, J**) Displacement of cell tracks in the 'mediolateral' (**I**) and 'anteroposterior' (**J**) dimensions in explants from *ndr2*-injected (blue) and control *RFP*-injected (gray) WT embryos (as in *Figure 1*). Each dot represents a single cell track, each color represents an individual explant. N = 4 explants of each condition from two independent trials. (**K**) Absolute displacement of cell tracks in the ML and AP dimensions. (**L**) Representative images of WISH for the transcripts indicated in explants from WT embryos injected with 10 pg *ndr2* RNA. Scale bars are 200 µm.

The online version of this article includes the following figure supplement(s) for figure 3:

**Figure supplement 1.** Nodal ligand levels regulate cell fate and extension of explants.

**Figure supplement 2.** Cell divisions within *ndr2*-expressing explants.

WISH assays revealed that explants expressed markers of endoderm, mesoderm (Spemann-Mangold organizer, axial and paraxial), and neuroectoderm according to the dose of *ndr2* with which they were injected, consistent with Nodal-dependent tissue induction in intact embryos (*Chen and Schier, 2001*; *Thisse et al., 2000*; *Feldman et al., 2000*; *Sampath et al., 1998*). For example, low doses of *ndr2* induced robust expression of the neuroectoderm marker *sox2* and some paraxial mesoderm (*tbx16*), whereas high doses induced expression of the endoderm marker *sox17* and the organizer gene *gsc*, but less neuroectoderm (*Figure 3—figure supplement 1*). *GFP-* or un-injected control explants expressed none of these tissue-specific markers at appreciable levels. Notably, explants injected with 10 pg *ndr2* exhibited discrete gene expression domains: the mesoderm markers *tbxta*, *tbx16*, and *noto* were nearly always restricted to one end, *sox17* was present in small spots (likely to be individual endoderm cells), and *sox2* was observed in a striped pattern along the long axis of each explant (*Figure 3L*, *Figure 3—figure supplement 1*). Together, these results demonstrate that Nodal signaling specifies a number of tissue types in discrete, spatially organized domains and promotes C and E morphogenesis to varying degrees depending on ligand dose within isolated naïve blastoderm.

## Blastoderm explants exhibit asymmetric Nodal signaling

The gene expression patterns observed in *ndr2*-expressing explants revealed asymmetry along the axis of extension (*Figure 3L*), which is known to be critical for C and E morphogenesis of *Xenopus* explants (*Ninomiya et al., 2004*). To test whether graded Nodal signaling activity could account for this asymmetry, we immuno-stained *ndr2*-injected and uninjected WT control explants for phosphorylated Smad2, an indicator of active Nodal signaling (*Figure 4A*; *Souchelnytskyi et al., 1997*). Using DAPI co-staining to create a nuclear mask, we used 3D automated object detection to quantify the location and pSmad2 staining intensity of all nuclei within each explant (*Figure 4A*, see 'Materials and methods'). After filtering out nuclei below a threshold background staining level, we compared the spatial distribution of the resulting pSmad2-positive nuclei along the 'axis' of each explant (*Figure 4B–C*). Comparing pSmad2+ nuclei (*Figure 4B–C*, blue dots) to all nuclei (gray dots) revealed in *ndr2*-injected explants a significant asymmetry of their distribution, which began at 6 hpf and increased until 8 hpf (p<0.0001, K-S tests), but very few pSmad2+ nuclei that exhibited little to no asymmetric distribution in uninjected controls (*Figure 4B, C*). Of note, the timing of pSmad2 detection differed between *ndr2*-injected explants and intact WT embryos, where an appreciable pSmad2 signal was detectable by 5 hpf (*Figure 4—figure supplement 1*), consistent with previous reports (*van Boxtel et al., 2015*; *Dubrulle et al., 2015*). pSmad2 then persisted in explants until at least 8 hpf, whereas no signal was detected after 6 hpf in embryos (*Figure 4—figure supplement 1*). No pSmad2 signal was detectable in embryos treated with the Nodal inhibitor SB-505124

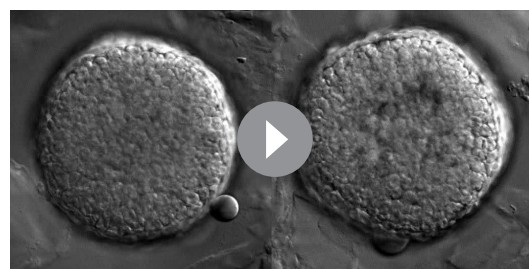

**Video 2.** Ex vivo extension of zebrafish blastoderm explants. Time-lapse differential interference contrast (DIC) series from 7 hpf to 12.5 hpf of representative explants from an uninjected WT embryo (left) and a WT embryo injected with 10 pg *ndr2* RNA (right).

https://elifesciences.org/articles/54445#video2

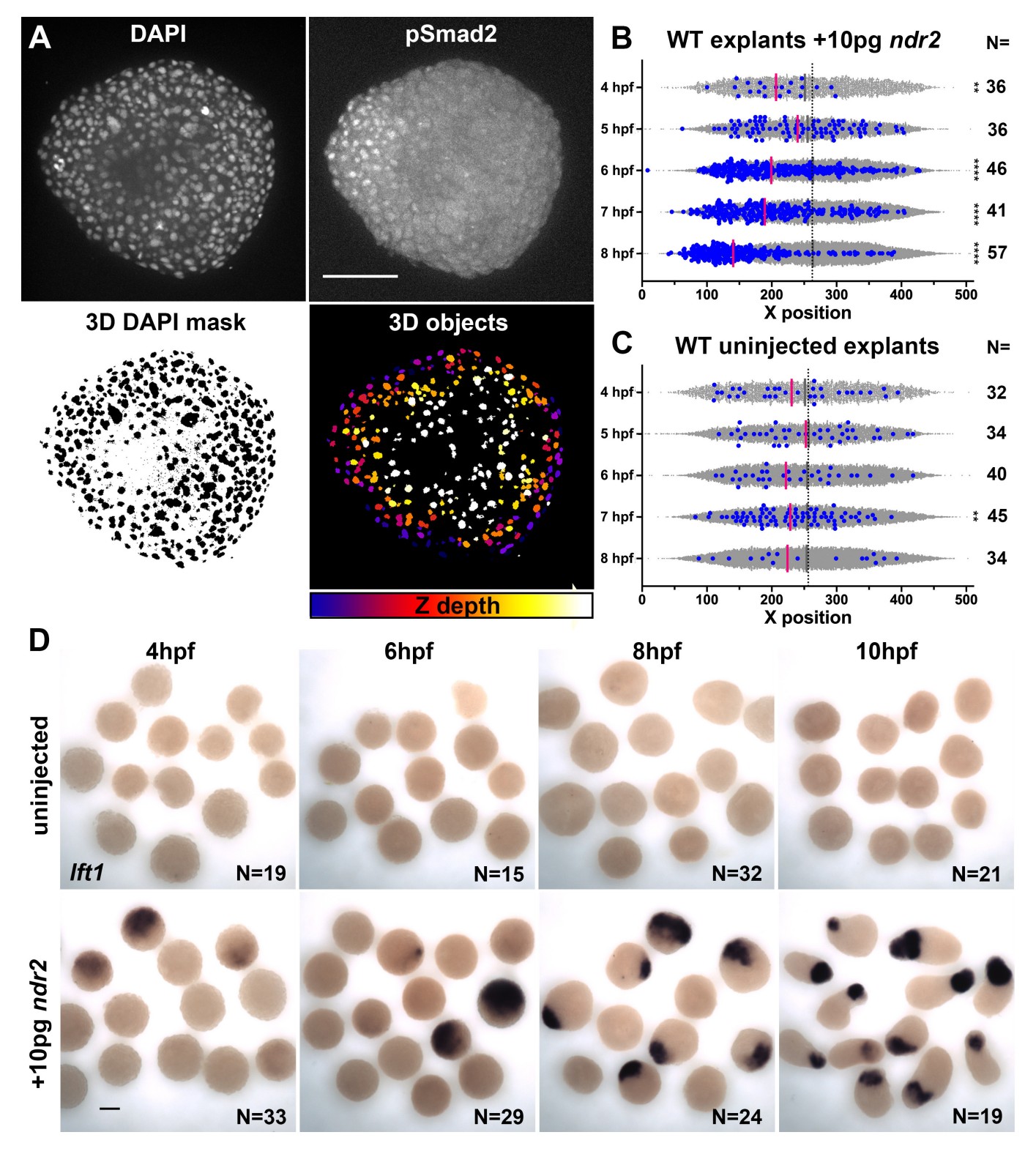

**Figure 4.** Nodal-expressing explants exhibit asymmetric Nodal signaling activity. (**A**) Representative confocal images of immunofluorescent staining for phosphorylated Smad2 and DAPI-labeled nuclei in 8hpf explants from WT embryos injected with 10 pg *ndr2*. DAPI z-stacks were used to create a three-dimensional mask from which nuclear pSmad intensities were detected and measured in an automated fashion. (**B, C**) Axis position of pSmad2-positive nuclei (blue) and all nuclei (gray) in explants from WT embryos injected with 10 pg *ndr2* (**B**) or uninjected (**C**) at the time points indicated. Each dot represents a single nucleus, pink bars are median values among pSmad2[+] nuclei. N indicates the number of explants in each condition from five

*Figure 4 continued on next page*

Figure 4 continued

independent trials. Kolmogorov-Smirnov tests were used to compare the distribution of pSmad$^+$ nuclei to all nuclei; ****, p<0.0001; **, p<0.01. (D) Representative images of WISH for *lefty1* in uninjected (top) and *ndr2*-injected (bottom) explants fixed at the time points indicated. Scale bars are 100 µm.

The online version of this article includes the following figure supplement(s) for figure 4:

**Figure supplement 1.** Nodal signaling activity in intact embryos.

(*DaCosta Byfield et al., 2004*; *Figure 4—figure supplement 1*), indicating that this antibody specifically detected Nodal signaling activity. Further evidence of asymmetric Nodal signaling within explants was provided by increasing levels and asymmetry of *lefty1* (*lft1*) expression, a negative feedback inhibitor and direct transcriptional target of Nodal signaling (*Meno et al., 1999*), in *ndr2*-expressing but not control explants (*Figure 4D*). These results demonstrate that injection of *ndr2* mRNA at the one-cell stage produces extending explants with asymmetric Nodal signaling activity.

## Nodal signaling and PCP promote cell polarity underlying C and E ex vivo

*Xenopus* animal cap explants that are exposed to Activin signaling exhibit robust ML cell polarization and intercalation (*Ninomiya et al., 2004*; *Shindo et al., 2008*). To analyze planar cell polarity underlying Nodal-driven C and E in zebrafish explants, we quantified cell alignment using live fluorescent membrane labeling (*Figure 5*). At the equivalent of the 2–4S stage, when intact WT embryos exhibit strong ML cell alignment (*Sepich and Solnica-Krezel, 2016*), cells within *GFP*-expressing control explants were randomly oriented (*Figure 5A,C* median angle = 44°). This was in stark contrast to *ndr2*-expressing explants, whose extension was accompanied by robust ML alignment (defined as perpendicular to the axis of extension) of cells (*Figure 5A, C* median angle = 19°). This result demonstrates that Nodal signaling is sufficient to induce ML cell alignment underlying C and E morphogenesis in populations of otherwise naïve embryonic zebrafish cells.

To ask whether Nodal-dependent ex vivo extension and ML cell alignment require PCP signaling, we generated explants from embryos that were co-injected with 10 pg *ndr2* mRNA and *vangl2* MO. *Vangl2* morphant explants exhibited overall lengths and length/width ratios that were significantly reduced compared with those in explants expressing *ndr2* alone, but significantly higher than those in uninjected controls (*Figure 5B, D, E*). It was recently reported that zebrafish blastoderm explants from embryos mutant for the PCP components *wnt11*, *wnt5b*, and *fzd7a/b* exhibited a similar reduction in ex vivo extension (*Schauer et al., 2020*). Live imaging of fluorescently labeled cell membranes further revealed that ML cell alignment was reduced but not entirely randomized in *vangl2* morphant explants compared with explants expressing *ndr2* alone (*Figure 5A, C* median angle = 26°). Because Nodal is necessary and sufficient for ML cell polarization ex vivo, and this polarity is reduced upon disruption of PCP signaling, these results indicate that PCP signaling functions downstream of Nodal in explant extension.

## Nodal signaling promotes ex vivo C and E independent of mesoderm

The requirement for Nodal signaling in zebrafish axis extension is well-described, but evidence suggests that this role is secondary to the ability of Nodal to specify mesoderm (*Aquilina-Beck et al., 2007*; *Gonsar et al., 2016*; *Araya et al., 2014*; *Smutny et al., 2017*). However, the presence of mesoderm in the absence of Nodal is not sufficient for C and E to occur (*Ninomiya et al., 2004*; *Howard and Smith, 1993*), and loss of a subset of *Xenopus* Nodal ligands disrupts C and E without affecting mesoderm formation (*Luxardi et al., 2010*), indicating a possible primary role for Nodal in extension morphogenesis. This is consistent with our results showing that polarity of MZ*oep*$^{-/-}$ donor cells is only partially restored by transplantation into WT hosts (*Figure 1*), which both corroborate the importance of mesoderm for proper C and E and suggest an additional cell-autonomous role for Nodal signaling in C and E cell behaviors. To determine whether Nodal plays a mesoderm-independent role in ex vivo C and E, we used SB-505124 (SB) to disrupt Nodal signaling within *ndr2*-expressing explants after mesoderm was specified (*Figure 6A*). Addition of 50 µM SB at 4, 5, or 6 hpf completely blocked both explant extension and expression of the mesoderm markers (and direct transcriptional targets of Nodal signaling; *Dubrulle et al., 2015*) *tbxta* and *noto* at 2–4S

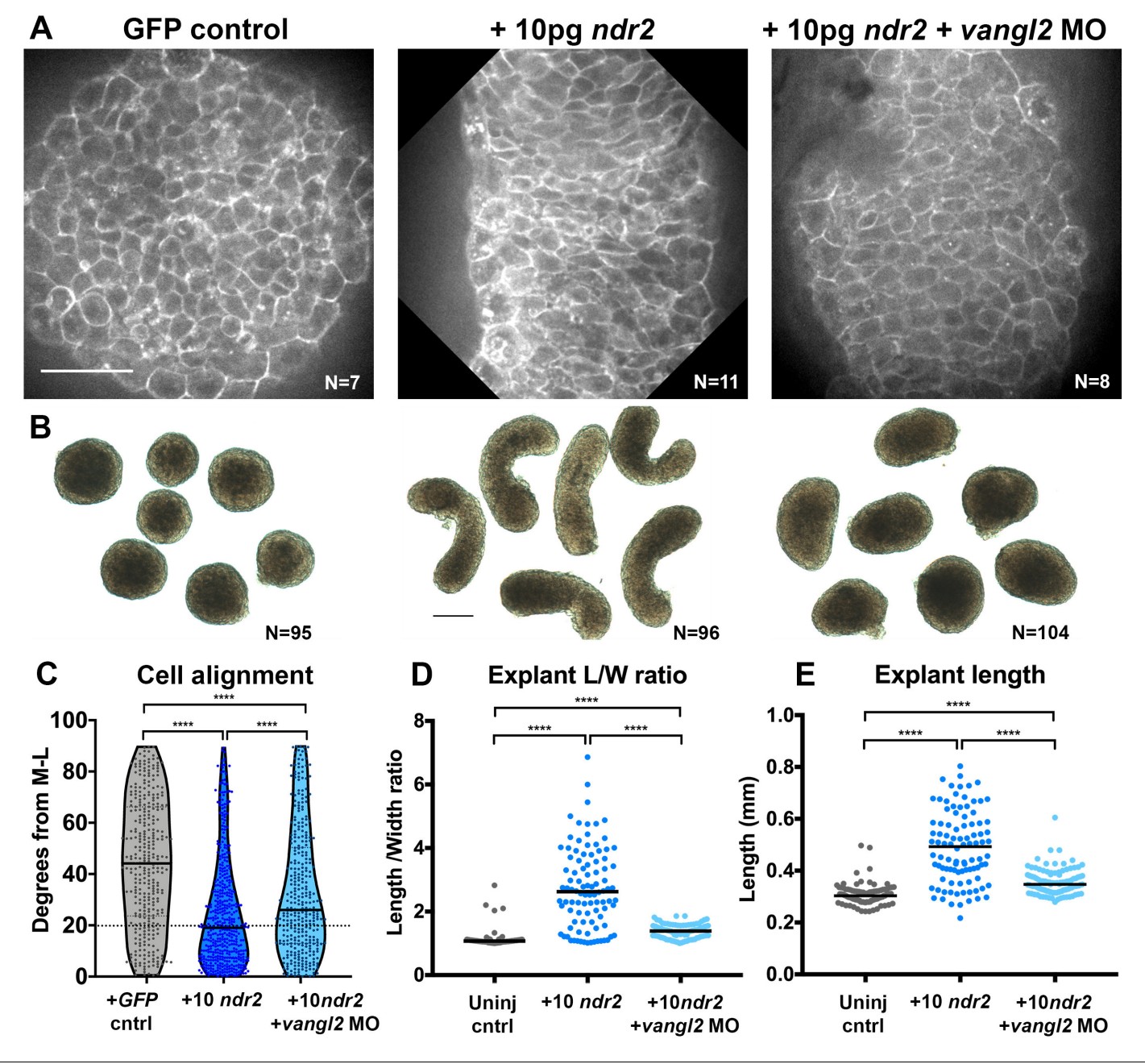

**Figure 5.** Disrupted PCP reduces Nodal-induced cell polarization and C and E within explants. (A) Representative confocal micrographs of live membrane-labeled explants of the indicated conditions at the equivalent of the 2–4 somite stage from two independent trials. (B) Representative bright-field images of blastoderm explants at 2–4S from four independent trials. (C) Explant cell alignment as in *Figure 1*. The number of explants in each condition is indicated on the corresponding panel in (A). Mediolateral (ML) is defined as orthogonal to the axis of extension. (D, E) Length/width ratios (D) and length (E) of explants depicted in (B). Each dot represents a single explant, black bars are median values. The number of explants in each condition is indicated on the corresponding panel in (B). p<0.0001; Kruskal-Wallis test. Scale bar is 50 µm in (A), 200 µm in (B).

stage (*Figure 6B–C*). By contrast, treatment of intact WT embryos with SB did not prevent AP axis extension after 5 hpf (*Figure 6—figure supplement 1*), consistent with previous reports (*Hagos and Dougan, 2007*). Critically, explants treated with SB at 8 hpf underwent extension but were significantly shorter than DMSO-treated controls despite robust mesoderm marker expression (*Figure 6B, C*) (p<0.0001, Mann-Whitney test), indicating that Nodal contributes to ex vivo C and E after mesoderm formation.

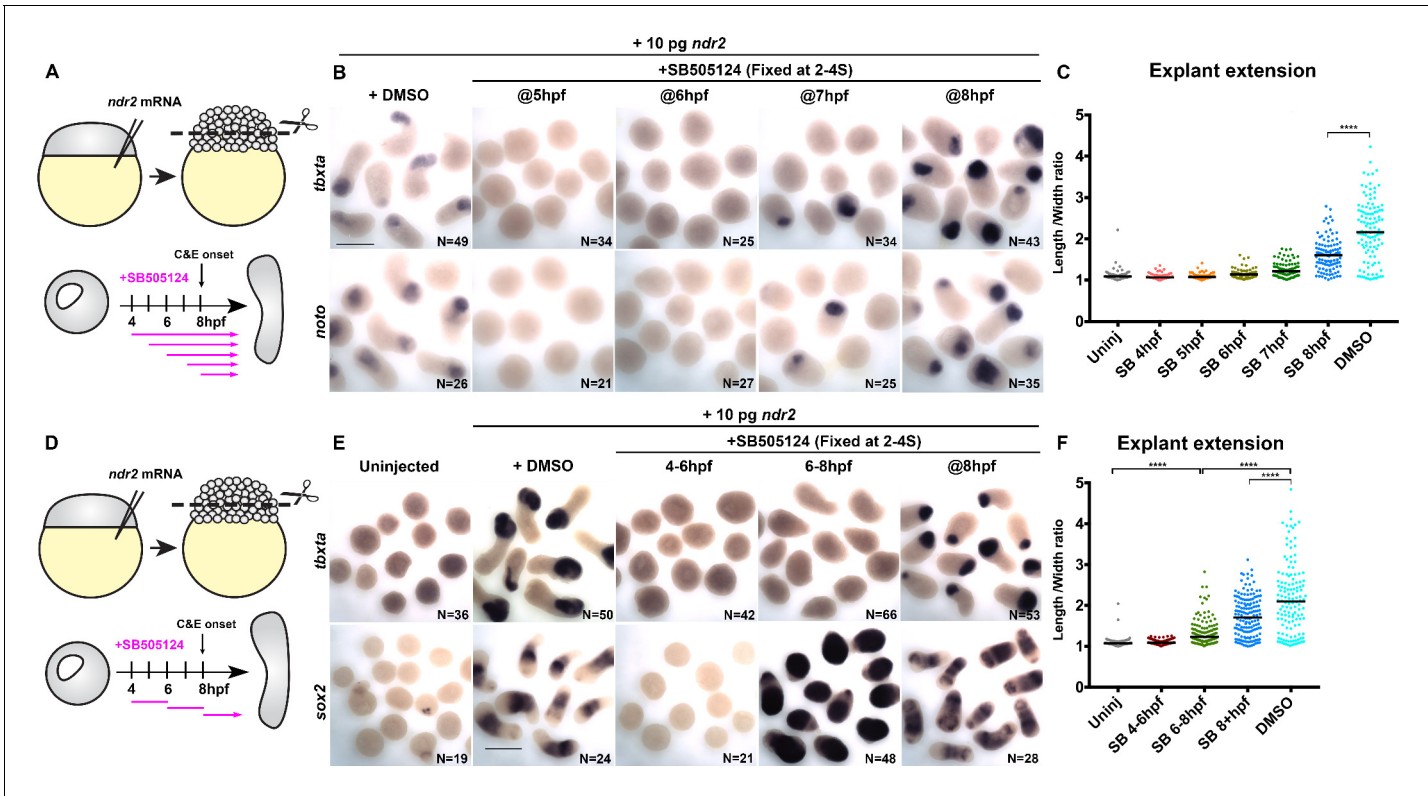

**Figure 6.** Nodal promotes ex vivo C and E independent of mesoderm. (**A**) Diagram of the time course of SB-505124 (SB) treatment of *ndr2*-expressing explants. (**B**) Representative images of WISH for the transcripts indicated in explants from WT embryos injected with 10 pg *ndr2* RNA, treated with SB at the indicated time points, and fixed at the equivalent of the 2–4 somite stage from four independent trials. (**C**) Length/width ratios of explants shown in (**B**). Each dot represents a single explant, black bars are median values; p<0.0001, Mann-Whitney test. (**D**) Diagram of the time course of SB treatment of *ndr2*-expressing explants followed by washout. (**E**) Representative images of WISH for the indicated transcripts in explants from WT embryos injected with 10 pg *ndr2* RNA, treated with SB at the indicated time points, and fixed at the equivalent of the 2–4 somite stage from four independent trials. (**F**) Length/width ratios of explants shown in (**E**), as in panel (**C**). ****p<0.0001, Mann-Whitney test. Scale bars are 300 μm.

The online version of this article includes the following figure supplement(s) for figure 6:

**Figure supplement 1.** Nodal inhibitor treatment of intact embryos.

**Figure supplement 2.** Time-course of Nodal inhibition in *ndr2*-expressing explants.

We next sought to evaluate the role of Nodal signaling in the absence of mesoderm by disrupting mesoderm formation in *ndr2*-expressing explants with a discrete two-hour pulse of SB, then allowing Nodal signaling to resume upon wash-out of the inhibitor (***Figure 6D***). Treatment from 4 hpf to 6 hpf completely blocked mesoderm marker expression and extension, even after the inhibitor was removed (***Figure 6E, F***). Treatment from 6 hpf to 8 hpf similarly blocked expression of the mesoderm markers *tbxta, noto*, and *tbx16*, but dramatically increased expression of the neuroectoderm markers *sox2* and *otx2b* by WISH (***Figure 6E***, ***Figure 6—figure supplement 2***). Moreover, these neural-only explants exhibited marked (albeit reduced compared with DMSO controls) extension (***Figure 6E, F***), demonstrating that Nodal signaling promotes ex vivo neuroectoderm C and E even in the absence of mesoderm. Because sustained SB treatment beginning at 6 hpf completely blocked explant extension (***Figure 6B–C***), this extension must be driven by Nodal signaling after removal of the inhibitor at 8 hpf. Indeed, inhibiting only this later phase of signaling by SB treatment at 8 hpf prevented full explant extension even in the presence of mesoderm (***Figure 6C, F***), indicating that Nodal signaling after C and E onset contributes significantly to ex vivo extension morphogenesis. These results support a tissue-autonomous requirement for Nodal signaling in ex vivo neuroectoderm C and E that is distinct from its role in mesoderm formation.

## Discussion

Coordination of embryonic patterning and morphogenesis is among the most fundamental and least understood problems in developmental biology. AP axial patterning is necessary for the evolutionarily conserved gastrulation movements of C and E within *Drosophila* (*Irvine and Wieschaus, 1994*; *Paré et al., 2014*) and vertebrate embryos (*Ninomiya et al., 2004*), but how C and E movements are coordinated with embryonic patterning in vertebrates is only beginning to be understood. The noncanonical Wnt/PCP signaling network is thought to act as a molecular compass that recognizes anterior and posterior cell edges in order to mediate ML cell polarization (*Gray et al., 2011*). Although exogenous Wnt ligands were shown to reorient PCP components and planar cell polarity in zebrafish and *Xenopus* gastrulae (*Lin et al., 2010*; *Chu and Sokol, 2016*), it is unclear whether endogenous Wnt ligands, either exclusively or in cooperation with additional signals, orient the PCP compass in vivo. Additional mechanical cues such as tension at tissue boundaries and directional strain also regulate planar cell polarity (*Chien et al., 2015*; *Williams et al., 2018*), but how such cues are linked to axial patterning is not well understood. Here, using a combination of intact zebrafish gastrulae and an ex vivo model of axial extension, we have defined critical roles for the morphogen Nodal in regulating cell behaviors underlying C and E of the primary embryonic axis. Our findings support a model in which Nodal signaling promotes C and E cell behaviors both upstream of and in parallel with PCP signaling, while additional Nodal-independent mechanisms — likely AP patterning cues — polarize PCP signaling (*Figure 7*).

Because Nodal signaling is necessary for both AP axial patterning (*Chen and Schier, 2001*; *Thisse et al., 2000*; *Feldman et al., 2000*) and polarized C and E cell behaviors (this work), it is a prime candidate to act upstream of PCP to orient the compass, thereby coordinating axial patterning and morphogenesis. Indeed, we found that impaired C and E in the neuroectoderm of Nodal signaling-deficient gastrulae was associated with reduced ML cell polarization (*Figure 1*), a phenotype also observed in PCP mutants (*Jessen et al., 2002*; *Kilian et al., 2003*; *Topczewski et al., 2001*; *Ulrich et al., 2003*). We further found, as others have observed for Activin and Nodal (*Ninomiya et al., 2004*; *Shindo et al., 2008*; *Xu et al., 2014*), that Nodal can induce both ML cell polarity and C and E in naïve explants (*Figure 3*). Moreover, this ex vivo Nodal-induced C and E is strongly reduced when PCP signaling is disrupted (*Figure 5*), indicating that Nodal functions upstream of PCP signaling. However, additional in vivo evidence suggests a more complex interaction between these two signaling systems. We found that within Nodal-deficient MZ*oep* mutant gastrulae, transcripts encoding PCP signaling components were expressed at largely normal levels (*Figure 2—figure supplement 1*) and the asymmetric intracellular localization of the core PCP component Prickle-GFP was only mildly affected (*Figure 2*). Finally, defects in C and E cell behaviors and axis extension, resulting from complete loss of Nodal signaling, were exacerbated by disrupted PCP signaling (*Figure 2*), providing additional evidence that PCP signaling remains active in the absence of Nodal. On the basis of these lines of evidence, we posit that Nodal is not absolutely required for the asymmetric distribution of this core PCP component nor for the transcriptional regulation of PCP genes in vivo, and plays only a minor role in regulating PCP signaling in intact embryos. Importantly, this indicates that Nodal is required for polarized C and E cell behaviors *in addition to* intact PCP signaling, rather than acting strictly upstream. Taken together, these data demonstrate that PCP and Nodal are each necessary – but not sufficient – for full ML polarization of cell behaviors underlying C and E, suggesting that Nodal functions largely in parallel with PCP in vivo (*Figure 7*). They also suggest that an additional signal (or signals) beyond Nodal instructs PCP signaling and the asymmetry of its components within the gastrula ('X' in *Figure 7*).

Results from explant experiments lead us to refined conclusions regarding the relationship between Nodal and PCP. Because 1) no cell polarity or C and E is observed in the absence of Nodal signaling in naïve blastoderm explants, 2) expression of Nodal ligands is sufficient to induce PCP-dependent ML cell polarization and C and E in the absence of other apparent patterning cues (*Figures 3* and *5*), and 3) this Nodal-induced ML cell polarization is strongly reduced by interference with PCP signaling (*Figure 5*), we interpret these results as indicating that PCP functions wholly downstream of Nodal in this ex vivo context (*Figure 7*). Although we cannot rule out the possibility that an additional non-Nodal signal regulates PCP in explants ('X'), any such signal would also operate downstream of Nodal. It is also possible that 'X' functions strictly in parallel with Nodal in vivo, and that this signal is absent from the explant system.

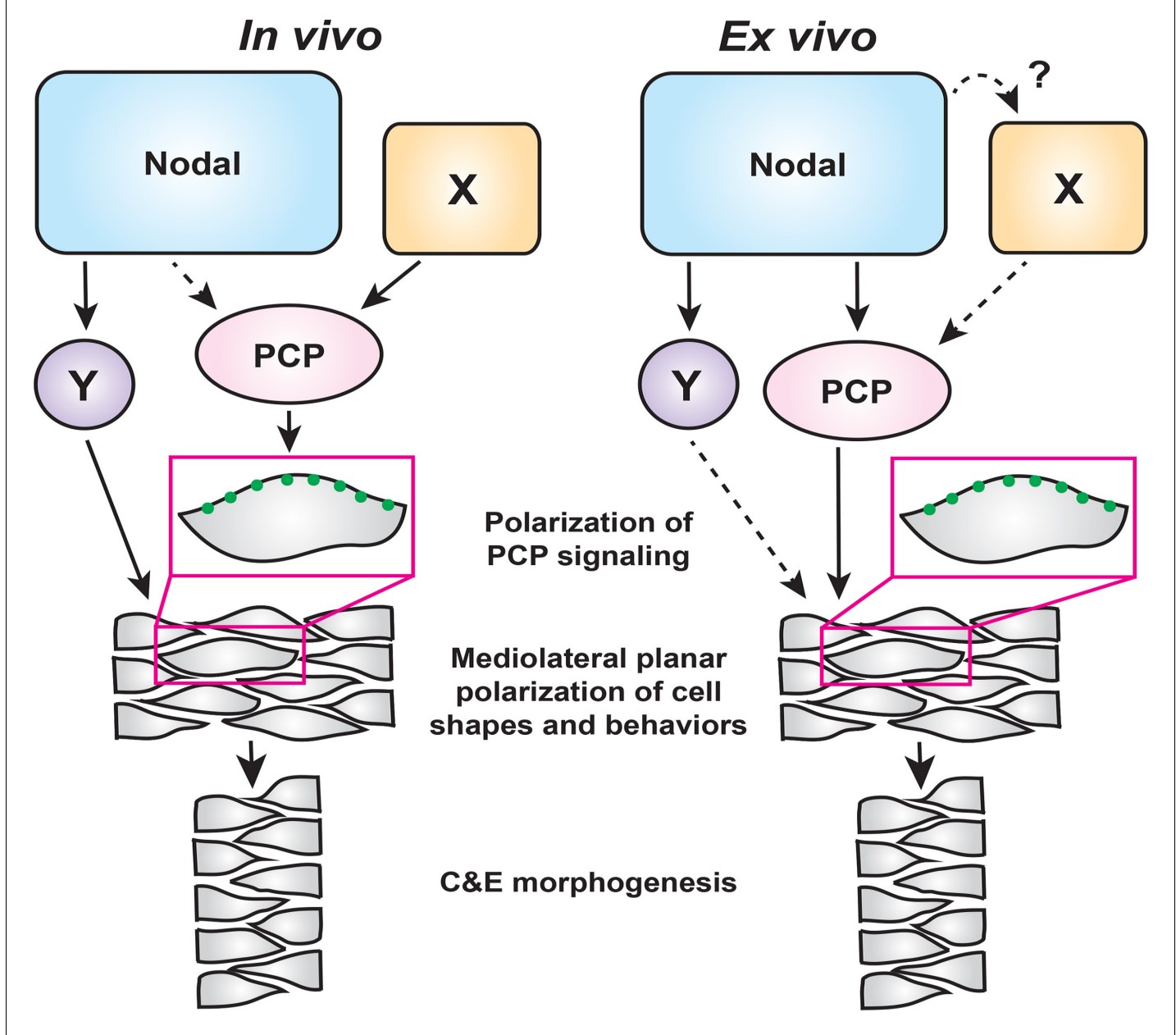

**Figure 7.** A model for the roles of PCP and Nodal signaling in C and E gastrulation movements. In intact embryos (left), Nodal signaling acts largely in parallel with PCP signaling to regulate the ML cell polarization that underlies C and E. PCP signaling activity and localization of its components are regulated by an additional unknown signal(s) (X), and maintains residual polarizing activity in the absence of Nodal. In embryonic explants (right), PCP signaling activity and C and E cell behaviors are regulated wholly downstream of Nodal signaling.

C and E and cell polarization were reduced but not completely abolished in *ndr2*-expressing *vangl2* morphant explants (*Figure 5*), as is observed in intact *trilobite/vangl2* mutant embryos (*Jessen et al., 2002*). This may indicate that Nodal contributes to polarized cell behaviors ex vivo through an additional PCP-independent mechanism ('Y' in *Figure 7*), that knockdown of *vangl2* alone does not completely abolish PCP signaling, or a combination of both. Indeed, accumulating evidence suggests that PCP is a complex signaling network comprised of multiple functionally discrete modules, such that loss of one may not entirely disrupt PCP as a whole (*Gray et al., 2011*). For

example, *vangl2* and *gpc4* compound zebrafish mutants have an additive C and E phenotype, imply-ing that these two PCP components do not work in a simple linear pathway (*Marlow et al., 1998*). *scribble* and *vangl2* were also recently reported to function in parallel with non-canonical Wnt ligands to align hair cells in zebrafish (*Navajas Acedo et al., 2019*), reinforcing the notion of multiple modules under the larger PCP umbrella. This complexity raises the possibility that Nodal signaling regulates only a subset of PCP signaling modules. For instance, although Nodal is not strictly required for asymmetric localization of Pk-GFP or Vangl2-dependent cell polarity in vivo (*Figure 2*), it may regulate the localization/activity of other PCP modules or PCP-associated molecules.

The results reported here are consistent with, but expand significantly upon, previous models for the role of Nodal in gastrulation morphogenesis. *Ninomiya et al., 2004* proposed that AP pattern-ing by Nodal-related signals is required to orient the polarity of PCP signaling during C and E gas-trulation movements. Although we similarly found that Nodal signaling is sufficient to induce both asymmetric patterning and PCP-dependent C and E, our genetic analyses indicate that Nodal is not strictly required for PCP polarization in vivo, and that the role of Nodal in C and E is not limited to AP patterning. Furthermore, although we observed asymmetric Nodal signaling in extending explants (*Figure 4*), it is not yet clear that this asymmetry is required for C and E cell behaviors. The source of this asymmetry is also unclear because the embryos were injected with *ndr2* RNA at the single-cell stage. We speculate that feed-back and feed-forward signaling mechanisms (*Müller et al., 2012*) may act to amplify small (and unavoidable) asymmetries in the distribution of injected RNAs. The importance of Nodal in axis extension has also been attributed to its ability to specify mesoderm (*Aquilina-Beck et al., 2007*; *Gonsar et al., 2016*; *Araya et al., 2014*), implying that Nodal-dependent mesoderm is required for C and E rather than Nodal signaling per se. We demonstrate here that Nodal contributes to ex vivo extension after mesoderm is specified and even in the absence of mesoderm (*Figure 6*), arguing for a primary, mesoderm-independent role for Nodal in C and E in addition to its well-described mesoderm-dependent functions. Transcriptional targets of Nodal signaling that regulate C and E independently of either PCP or mesoderm forma-tion ('Y' in *Figure 7*) remain largely unknown, and identification of these molecules will be an impor-tant goal for future studies.

The blastoderm explants described in this study provide a robust, simplified model of axial exten-sion in which a signaling molecule of interest — in this case, Nodal — can be studied independently of endogenous patterning and signaling events. It has been shown that some methods of zebrafish blastoderm explantation do not require the addition of Nodal or any other signaling molecules to induce extension (*Schauer et al., 2020*; *Trivedi et al., 2019*), but these explants are generated from the entire blastoderm and therefore contain signals (Nodal and otherwise) already present at the embryonic margin (*Erter et al., 1998*). By contrast, unmanipulated explants containing only the ani-mal-most blastoderm are comparatively naïve, as they exclude endogenous signals from the margin, and fail to extend (*Xu et al., 2014*; *Sagerström et al., 1996*; *Schauer et al., 2020*; *Trivedi et al., 2019* [this work]), similar to classic *Xenopus* animal cap explants (*Howard and Smith, 1993*). Although explants from any species are a powerful tool, we must acknowledge ways in which this system differs from intact embryos. Namely, because explants do not contain the full complement of molecular signals and tissue interactions present in intact embryos, the contribution of additional sig-nals to C and E may be masked by the reliance of explant extension on Nodal alone. This is illus-trated by the function of PCP both downstream of and in parallel with Nodal in vivo, but strictly downstream of Nodal ex vivo (*Figure 7*). The absence of a yolk cell also dramatically alters the geometry of explanted tissues and removes signaling input from the extraembryonic yolk syncytial layer (*Schauer et al., 2020*). Despite these differences, explants exhibit a suite of complex, biologi-cally relevant behaviors in common with intact embryos, including C and E morphogenetic move-ments, ML cell polarization, timing of C and E onset, and transcriptional responses to Nodal. These explants therefore provide a simplified platform that has allowed for new insights into the role of Nodal signaling in C and E morphogenesis and for dissection of its complex relationship with PCP. Although Nodal signaling functions wholly upstream of PCP-dependent ML planar polarity of cells ex vivo, in vivo it functions in an overlapping fashion and cooperates with PCP signaling, whose activity is regulated by additional, as yet unidentified, signaling events (*Figure 7*).

# Materials and methods

## Key resources table

| Reagent type (species) or resource | Designation | Source or reference | Identifiers | Additional information |
|---|---|---|---|---|
| Gene (*Danio rerio*) | *tdgf1* (*oep*) | ZFIN | RRID:ZFIN_ZDB-GENE-990415-198 | |
| Gene (*Danio rerio*) | *ndr2* (*cyc*) | ZFIN | RRID:ZFIN_ZDB-GENE-990415-181 | |
| Strain, strain background (*Danio rerio*) | AB* | ZIRC | RRID:ZFIN_ZDB-GENO-960809-7 | |
| Genetic reagent (*Danio rerio*) | *oep^{tz257}* | *Hammerschmidt et al., 1996* | RRID:ZFIN_ZDB-GENO-130130-2 | Point mutation |
| Recombinant DNA reagent | pJZoepFlag1-2 in pcDNA3 (plasmid) | *Zhang et al., 1998* | | Template for in vitro transcription |
| Recombinant DNA reagent | *ndr2* in PCS2+ (plasmid) | *Sampath et al., 1998* | | Template for in vitro transcription |
| Recombinant DNA reagent | *membrane Cherry* in PCS2+ (plasmid) | Gift from Dr Fang Lin | | Template for in vitro transcription |
| Recombinant DNA reagent | *membrane eGFP* in PCS2+ (plasmid) | *Wallingford and Harland, 2002* | | Template for in vitro transcription |
| Recombinant DNA reagent | *H2B-RFP* in PCS2 (plasmid) | Gift from Dr John Wallingford | | Template for in vitro transcription |
| Recombinant DNA reagent | *Drosophila* Prickle-GFP (*plasmid*) | *Jenny et al., 2003* | | Template for in vitro transcription |
| Antibody | Anti-phospho Smad2/3 | Cell Signaling Technology #8828 | RRID:AB_2631089 | IF (1:1000) |
| Antibody | Invitrogen AlexaFluor 488 goat anti-rabbit IgG | Thermo Fisher #A-11008 | RRID:AB_143165 | IF (1:1000) |
| Antibody | Roche Anti-digoxigenin-AP Fab fragments | Millipore Sigma #11093274910 | RRID:AB_2734716 | 1:5000 |
| Commercial assay or kit | Roche Digoxigenin RNA labelling mix | Millipore Sigma #11277073910 | | |
| Other | Roche BM Purple AP staining solution | Millipore Sigma #11442074001 | | |
| Sequenced-based reagent | MO4-*vangl2* Morpholino antisense oligonucleotide | GeneTools (*Williams et al., 2012*) | | AGTTCCACCTTACTCCTGAGAGAAT |
| Commercial assay or kit | Invitrogen mMessage mMachine SP6 kit | Thermo Fisher # AM1340 | | |
| Commercial assay or kit | RNeasy Mini kit | Qiagen #74104 | | |
| Chemical compound, drug | Ambion Trizol reagent | Thermo Fisher #15596018 | | |
| Chemical compound, drug | SB-505124 | Millipore Sigma # S4696 | | 50 mM |
| Peptide, recombinant protein | Roche Pronase | Millipore Sigma #10165921001 | | |
| Other | New-born calf serum | Invitrogen #26010–066 | | |

*Continued on next page*

*Continued*

| Reagent type (species) or resource | Designation | Source or reference | Identifiers | Additional information |
|---|---|---|---|---|
| Software, algorithm | Imaris | Oxford Instruments | RRID:SCR_007370 | Live cell tracking |
| Software, algorithm | ImageJ/FIJI | ImageJ/FIJI | RRID:SCR_002285 | Image analysis |
| Software, algorithm | Prism 8 | Graphpad | RRID:SCR_002798 | Statistics and graphs |

## Zebrafish

Adult zebrafish were raised and maintained according to established methods (*Westerfield, 1993*) in compliance with standards established by the Washington University Institutional Animal Care and Use Committee. Embryos were obtained from natural matings and staged according to morphology as described (*Kimmel et al., 1995*). All studies on WT embryos were carried out in AB* backgrounds. Additional lines used include *oep^tz257* (*Hammerschmidt et al., 1996*) on AB* background. *oep^−/−* embryos were rescued by injection of 50 pg synthetic *oep* RNA (*Zhang et al., 1998*) and raised to adulthood, then intercrossed to generate maternal-zygotic *oep^−/−* embryos. Fish were chosen from their home tank to be crossed at random, and the resulting embryos were also chosen from the dish at random for injection and inclusion in experiments.

## Microinjection of synthetic RNA and morpholino oligonucleotides

Single-celled embryos were aligned within agarose troughs made from custom plastic molds and injected with 1–3 nL volumes using pulled glass needles. Synthetic mRNAs for injection were made by in vitro transcription from linearized plasmid DNA templates using Invitrogen mMessage mMachine kits. Doses of RNA per embryo were as follows: 100 pg *membrane Cherry* (a kind gift from Dr Fang Lin), 50 pg membrane *eGFP* (*Wallingford and Harland, 2002*), 25 pg *H2B-RFP* (a kind gift from Dr. John Wallingford), 50 pg *Drosophila pk-gfp* (*Jenny et al., 2003*), and 2.5–100 pg *ndr2* (*Sampath et al., 1998*). Injection of 2 ng MO4-*tri/vangl2* (*Williams et al., 2012*) was carried out as for synthetic RNA.

## Pharmacological treatments

50 µM SB-505124 (Sigma #S4696) was added to the media of embryos and explants in agarose-coated 6-well plates at the times specified. For wash-out experiments, SB-containing medium was removed and explants were washed twice with 0.3x Danieau solution, before fresh explant medium was introduced.

## Immunofluorescent staining

Embryos and explants were stained for phosphorylated Smad2 as described in *van Boxtel et al., 2015*. Briefly, samples were fixed overnight in 4% paraformaldehyde (PFA), rinsed in phosphate-buffered saline (PBS) + 0.1% Tween-20 (PBT), and dehydrated to 100% methanol. Prior to staining, samples were rehydrated into PBS, rinsed in PBS + 1% Triton X-100, and incubated in ice-cold acetone at −20°C for 20 min. Samples were then blocked in PBS+ 10% FBS and 1% Triton X-100, and then incubated overnight at 4°C with an anti-pSmad2/3 antibody (Cell Signaling #8828) at 1:1000 in block. Samples were rinsed in PBT/1% Triton X-100 and incubated with Alexa Fluor 488 anti-Rabbit IgG (Invitrogen) at 1:1000. Embryos were co-stained with 4′,6-diamidino-2-phenylindole, dihydrochloride (DAPI) and rinsed in PBS + 1% Triton X-100 prior to mounting in 2% methylcellulose for confocal imaging.

## Whole mount in situ hybridization

Antisense riboprobes were transcribed using NEB T7 or T3 RNA polymerase and labeled with digoxygenin (DIG) (Roche). Whole-mount in situ hybridization (WISH) was performed according to *Thisse and Thisse, 2008*. Briefly, embryos and explants were fixed overnight in 4% PFA in PBS, rinsed in PBT, and dehydrated into methanol. Samples were then rehydrated into PBT, incubated for at least two hours in hybridization solution with 50% formamide (in 0.75 M sodium chloride, 75 mM

sodium citrate, 0.1% tween 20, 50 µg/mL heparin (Sigma), and 200 µg/mL tRNA) at 70°C, then hybridized overnight at 70°C with antisense probes diluted approximately 1 ng/µL in hybridization solution. Samples were washed gradually into 2X SSC buffer (0.3 M sodium chloride, 30 mM sodium citrate), and then gradually from SSC to PBT. Samples were blocked at room temperature for several hours in PBT with 2% goat serum and 2 mg/mL bovine serum albumin (BSA), then incubated over-night at 4°C with anti-DIG antibody (Roche #11093274910) at 1:5000 in block. Samples were rinsed extensively in PBT, and then in staining buffer (PBT +100 mM Tris [pH 9.5], 50 mM $MgCl_2$, and 100 mM NaCl) prior to staining with BM Purple AP staining solution (Roche).

## RNA-sequencing

RNA for sequencing was isolated from 50 pooled WT or MZ$oep^{-/-}$ embryos at the 90% epiboly stage from three independent clutches per genotype (three biological replicates). Embryos were lysed and total RNA was isolated using Trizol reagent (Ambion), then cleaned up using a Qiagen RNeasy kit. Samples were submitted to the Washington University Genome Technology Access Center for library preparation, including depletion of ribosomal RNA. Libraries were sequenced using an Illumina HiSeq3000 to obtain single-ended 50-bp reads. Raw reads were trimmed with cutadapt to remove low-quality bases and aligned to *Danio rerio* genome GRCz10 using STAR_2.5.4b (*Dobin et al., 2013*) with Ensembl v83 annotation. Aligned reads were quantified using feature-Counts 1.6.3 from the subreads package (*Liao et al., 2013*).

## Blastoderm explants

Embryos were injected with *ndr2, H2B-RFP,* and/or *membrane GFP* RNA (and MOs) at the one-cell stage as described above, or left uninjected, then dechorionated using Pronase (Roche). At the 256–512 cell stage, watchmaker's forceps were used to excise the animal portion of each embryo in an agarose-coated dish filled with 3X Danieau solution, cutting the blastoderm at approximately 50% of its height and collecting the animal-most ~1/3 of cells from each embryo, as described in *Xu et al., 2014*. Explants were allowed to heal briefly, then transferred into agarose-coated 6-well plates containing explant medium — comprised of Dulbecco's modified eagle medium with nutrient mixture F-12 (Gibco 11330032) containing 2.5 mM L-glutamine, 15 mM HEPES, 3% newborn calf serum (Invitrogen 26010–066), 50 units/mL penicillin, and 50 µg/mL streptomycin (10,000 U/mL pen-strep at 1:200, Gibco 15140163) — and incubated at 28.5°C until the desired stage.

## Transplantation

For cell autonomy and Pk-GFP transplants, host and donor embryos were injected with RNA encoding fluorescent nuclear and membrane markers, MOs, and/or Pk-GFP as described above. For cell autonomy, donor embryos were injected with *membrane Cherry* or *membrane GFP* and cells were transplanted into unlabeled hosts. For Pk-GFP localization, donor embryos were co-injected with *mCherry, prickle-GFP,* and *H2B-RFP* RNA and cells were transplanted into *mCherry*-injected hosts. Hosts and donors were dechorionated using Pronase, then arranged within the wells of a custom-molded agarose plate at approximately sphere stage. Approximately 20–40 cells were transferred from donors to hosts using a fine-pulled glass capillary.

## Microscopy

Live embryos/explants expressing fluorescent proteins were mounted in 0.75% low-melt agarose, and fixed embryos/explants subjected to immunofluorescent staining were mounted in 2% methylcellulose in glass- bottomed 35 mm Petri dishes for imaging using a modified Olympus IX81 inverted spinning disc confocal microscope equipped with Voltran and Cobolt steady-state lasers and a Hamamatsu ImagEM EM CCD digital camera. For live time-lapse series, 60–100 µm z-stacks with a 1–2 µm step were collected every 3–5 minutes (depending on the experiment) for 3 hours using a 20x or 40x dry objective lens for intact embryos and a 20x objective for explants. Temperature was maintained at 28.5°C during imaging using a Live Cell Instrument stage heater. For immuno-stained embryos and explants, 100 µm z-stacks with a 1 or 2 µm step were collected using a 10x or 20x dry objective lens, depending on the experiment. Bright-field and transmitted light images of live embryos and in situ hybridizations were collected using a Nikon AZ100 macroscope.

## Image analysis

ImageJ was used to visualize and manipulate all microscopy data sets.

### Individual cell measurements

Within live embryos and explants expressing either membrane eGFP or membrane Cherry, a single z-plane (in ubiquitously labeled embryos) or a projection of several z-planes (when measuring transplanted cells) through the neuroectoderm was chosen for each time point. To measure cell orientation and elongation, the AP axis in all embryo images was aligned prior to manual outlining of cells. A fit ellipse was used to measure the orientation of each cell's major axis and its aspect ratio. To assess Pk-GFP localization, isolated donor cells co-expressing Pk-GFP and H2B-RFP were scored according to the subcellular localization of GFP signal. Cell protrusions were manually detected within time-lapse stacks of transplanted neuroectoderm cells and their orientation measured in ImageJ. These orientations were binned into one of 12 sectors (30° each), which were categorized as mediolateral (within 60° of horizontal), anteroposterior (within 60° of vertical), or neither (the remaining 120°). The distribution of protrusions within each of these categories was then compared between experimental conditions using Chi-square tests.

### Automated nuclear tracking

Imaris software and the ImageJ TrackMate plugin were used to detect and track labeled nuclei automatically in the dorsal hemisphere of WT and MZ*oep* mutant gastrulae and in embryonic explants injected with RNA encoding H2B-RFP, and to produce color-coded depictions of their trajectories and measurements of speed and displacement. Track displacement in the X (mediolateral) and Y (anteroposterior) dimensions were calculated independently, and plotted against their starting positions within the embryo/explant using Graphpad Prism 8 software. Cell divisions were manually detected within time-lapse stacks of explants from WT embryos co-injected with *H2B-RFP* and *ndr2* RNA, and their locations with respect to the center of each explant were measured using ImageJ.

### Morphometric measurements

To measure the length/width ratios of explants, we divided the length of a segmented line drawn along the midline of each explant (accounting for curvature) by the length of a perpendicular line spanning the width of the explant near its midpoint. To measure width of the neural plate in whole-mount embryos, dorsal-view images were collected of each embryo, and a line was drawn from one side of the *dlx3b* expression domain to the other side at the level of the future mid-hindbrain boundary marked by *egr2b* expression. Length measurements were made similarly by measuring from the anterior to posterior aspects of the *dlx3b* expression domain in lateral-view images. Images were coded and analyses were performed blinded to ensure unbiased measurements.

### pSmad2 immunostaining in explants

DAPI z-stacks were converted into 3D masks, which were used to create z-stacks of nuclear pSmad2 labeling. All stacks were oriented such that the highest apparent pSmad2 signal (if any) was to the left, and the ImageJ '3D Objects counter' plugin was used to detect the location and pSmad2 fluorescence intensity of all nuclei in a given explant. All nuclei with fluorescence intensities above a threshold background level were categorized as pSmad2-positive.

## Statistical analysis

Graphpad Prism 8 software was used to perform statistical analyses and to generate graphs of data collected from embryo and explant images. The statistical tests used varied as was appropriate for each experiment and are described in the text and figure legends. Data were tested for normal distribution, and non-parametric tests (Mann-Whitney and Kolmogorov-Smirnov) were used for all non-normally distributed data. Normally distributed data with similar variance between groups were analyzed using parametric tests (T-tests). All tests used were two-tailed. Circular histograms were created using PAST software.

## Acknowledgements

We thank the Washington University Genome Technology Access Center for library preparation and sequencing services, the Washington University Center for Cellular Imaging for use of Imaris work stations, Dr Paul Gontarz for help with RNA-seq analysis, and Drs Alex Schier (and lab members), Diane Sepich, Ann Sutherland, Ray Keller, and Dave Shook for helpful discussions. This work was supported by National Institutes of Health awards K99HD091386 to MLKW and 1R35GM118179 to LS-K.

## Additional information

### Funding

| Funder | Grant reference number | Author |
| --- | --- | --- |
| National Institutes of Health | K99HD091386 | Margot L K Williams |
| National Institute of General Medical Sciences | R35GM118179 | Lilianna Solnica-Krezel |

The funders had no role in study design, data collection and interpretation, or the decision to submit the work for publication.

### Author contributions

Margot LK Williams, Conceptualization, Formal analysis, Funding acquisition, Investigation, Writing - original draft, Writing - review and editing; Lilianna Solnica-Krezel, Conceptualization, Funding acquisition, Writing - review and editing

### Author ORCIDs

Margot LK Williams https://orcid.org/0000-0001-9704-6301

### Ethics

Animal experimentation: This study was performed in strict accordance with recommendations in the Guide for the Care and Use of Laboratory Animals of the National Institutes of Health. Adult zebrafish were raised and maintained according to established methods and in compliance with standards established by the Washington University Animal Care and Use Committee (IACUC), approval number 20160116; Animal Welfare Assurance number A-3381-01.

### Decision letter and Author response

Decision letter https://doi.org/10.7554/eLife.54445.sa1
Author response https://doi.org/10.7554/eLife.54445.sa2

## Additional files

### Supplementary files

• Source data 1. Differentially expressed genes between WT and MZ*oep*−/− gastrulae detected by RNA-seq at the 90% epiboly stage. Values shown in columns H through M are counts per million reads mapped (CPM) for each biological replicate.

• Source data 2. CPM values for all genes within WT and MZ*oep*−/− gastrulae detected by RNA-seq at the 90% epiboly stage.

• Transparent reporting form

### Data availability

Sequencing data have been deposited in GEO under accession code GSE147302. Processed RNA-seq data have been provided in Source Data Files 1 and 2.

The following dataset was generated:

| Author(s) | Year | Dataset title | Dataset URL | Database and Identifier |
|---|---|---|---|---|
| Williams MLK, Solnica-Krezel L | 2020 | Differential gene expression in WT v. MZoep-/- zebrafish gastrulae | https://www.ncbi.nlm.nih.gov/geo/query/acc.cgi?acc=GSE147302 | NCBI Gene Expression Omnibus, GSE147302 |

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
