## [Decision Letter]

**Acceptance summary:**

The formation of the vertebrate anterior-posterior axis has long been known to depend on two major signaling pathways: Nodal controls tissue specification, whereas Planar Cell Polarity (PCP) signaling controls morphogenesis by regulating cell movements that elongate the anterior-posterior axis through convergent extension. This paper now provides strong evidence that both Nodal and PCP signaling participate in convergent extension. The authors found that Nodal signaling is necessary and sufficient to drive axis extension in zebrafish embryos and naive cell clumps, likely independently of its role in tissue specification. Together, these results support a revised model for the role of Nodal signaling in anterior-posterior axis formation.

**Decision letter after peer review:**

Thank you for submitting your article "Nodal and Planar Cell Polarity signaling cooperate to regulate zebrafish convergence & extension gastrulation movements" for consideration by *eLife*. Your article has been reviewed by two peer reviewers, one of whom is a member of our Board of Reviewing Editors, and the evaluation has been overseen by Didier Stainier as the Senior Editor. The reviewers have opted to remain anonymous.

The reviewers have discussed the reviews with one another and the Reviewing Editor has drafted this decision to help you prepare a revised submission.

Summary:

In this manuscript, Williams and Solnica-Krezel propose that Nodal and PCP signaling cooperate to regulate convergence and extension (CE) movements during zebrafish axis formation. The authors nicely show that the absence of Nodal signaling impairs extension and, to a lesser degree, also convergence movements of ectoderm. Moreover, they show that the cell alignments defects in Nodal signaling-deficient embryos are worsened by impaired PCP signaling. Similarly, the extension of blastoderm explants depends on the right dosage of Nodal signaling and is impaired in PCP signaling-deficient explants. The authors suggest that these results indicate that Nodal works both in parallel and upstream of PCP signaling in vivo. However, more evidence is needed to substantiate these claims.

Essential revisions:

1) The fact that the MZoep phenotype is worsened by the absence of Vangl2 indicates that both pathways are needed for ectoderm CE. Yet, it is not clear to what extent Nodal might also function upstream of PCP signaling. Especially, the evidence that PCP signaling is partially disrupted in MZoep mutants is not entirely convincing. To strengthen this conclusion, it would be good to test, for instance, (i) to what extent the MZoep CE phenotype can be rescued by PCP activation, and (ii) which aspect of PCP signaling (dynamic subcellular localization, expression level, turnover etc.) might partially depend on Nodal signaling.

2) If CE is affected by loss of Nodal activity, a widening of the neural plate would be expected similar to the findings for the *vangl2* MO. However, MZ*oep* embryos actually have narrower neural plates than WT (Figure 2E, F), undermining the proposal that Nodal regulates CE. Also, why are the markers in Figure 2E more intensely stained in MZ*oep* embryos compared to WT?

3) The statement "Orientation of MZ*oep*-/- cellular protrusions was also only partially improved within WT hosts (Figure 1I)" is not supported and should be adjusted, since the graphs in Figure 1H and I look similar. The finding that MZ*oep* cells transplanted into wildtype hosts display nearly normal protrusion orientation is interesting. Does this depend on the polarization of the surrounding cells and would, for instance, wildtype ectoderm cells have a more random orientation of the protrusions when transplanted into MZ*oep*?

4) Using the blastoderm explants, the authors suggest that convergence movements in this ex vivo system depend on Nodal signaling and that Nodal acts upstream of PCP signaling. However, the extension of the explant or the alignments of the cells aren't necessarily a readout of convergence movements. To claim this, it would be necessary to show actual convergence movements of the cells. Is it possible that elongation is promoted by a general increase in cell movements or differential proliferation? Is Prickle-GFP localization polarized in explants? Furthermore, injection of Nodal ligands also induces the specification of endoderm and mesoderm, which are both lacking in the uninjected control. The activation of PCP signaling during the extension of the explant could be due to the presence of endoderm and mesoderm rather than due to the activity of Nodal signaling per se. Drug treatment to block Nodal signaling after acquisition of cell fate could be a possible experiment to discriminate whether continuous Nodal signaling activity is needed. The paper would also benefit from a clearer distinction between elongation and CE. This is crucial since the effect of Nodal could in principle be mostly on prechordal plate migration rather than CE.

5) Nodal acts highly locally, not in a widespread manner throughout the embryo (see Figure 4—figure supplement 1 and Xu et al., 2014). How would Nodal regulate cell behavior far away in the neuroectoderm?

6) The conclusions rely on "significant differences" between experimental conditions determined by statistical tests. However, the effect sizes are often very small, and the variance within experimental groups is high. The low p values are a consequence of the very high sample numbers, but the biological significance of the determined effect sizes remains unclear. The authors should therefore apply bootstrapping or resampling methods using subsets from the data distributions to test whether the effect sizes are reproducible.

7) It is unclear why embryos uniformly injected with *ndr2* give rise to polarized explants. It would be good to show that the presence of asymmetry does not depend on asymmetric injection of Nodal ligands, e.g. by executing in situ hybridization stainings with probes for injected *ndr2* (perhaps with a GFP tag or the like to distinguish it from endogenously produced Nodals). If injected *ndr2* is uniformly distributed, is Nodal autoactivation required for polarization, i.e. what happens in MZ*sqt*;*cyc* embryos?

8) The authors state that "Although the PCP genes *wnt5b*, *vangl2* (*trilobite*), and *gpc4* (*knypek*) were expressed at apparently normal levels in MZ*oep*-/- gastrulae by whole mount in situ hybridization (WISH) (Figure 2—figure supplement 1B)", but the data does not show this; instead all genes seem to be upregulated in MZ*oep* embryos. Please adjust the conclusions.

---

## [Author Response]

Essential revisions:1) The fact that the MZoep phenotype is worsened by the absence of Vangl2 indicates that both pathways are needed for ectoderm CE. Yet, it is not clear to what extent Nodal might also function upstream of PCP signaling. Especially, the evidence that PCP signaling is partially disrupted in MZoep mutants is not entirely convincing. To strengthen this conclusion, it would be good to test, for instance, (i) to what extent the MZoep CE phenotype can be rescued by PCP activation, and (ii) which aspect of PCP signaling (dynamic subcellular localization, expression level, turnover etc.) might partially depend on Nodal signaling.

Based on our studies of MZ*oep* mutants we propose that PCP signaling is partially active in the absence of Nodal signaling and that the two pathways work largely in parallel in vivo. However, two pieces of data indicate that PCP signaling is somewhat perturbed in MZ*oep*-/- gastrulae. First, whereas we show that WT and mutant cells exhibit comparable enrichment of Prickle-GFP at their anterior cell membranes, we also note that the mutant cells exhibit an increased proportion of Pk-GFP puncta that are not limited to anterior cell edges compared with WT (Figure 2A). This has proven to be our most reliable read-out of PCP signaling status during gastrulation, and demonstrates a mild PCP phenotype in Nodal-deficient mutants. Second, we have included new RNA-sequencing data directly comparing gene expression between stage matched MZ*oep*-/- and WT embryos at 90% epiboly stage. Analysis of these data demonstrates that the expression of most PCP genes is unchanged in MZ*oep*-/- compared to WTs, but the core PCP component *prickle1b* is significantly reduced in these mutants, again demonstrating a mild effect on PCP function. These data have been included as the new panel A in Figure 2—figure supplement 1 and are described within the Results section.

Processed RNA-sequencing data are included as Source Data Files 1 and 2, and raw sequence data have been deposited within the NCBI Gene Expression Omnibus under GEO Accession Number: GSE147302. These data can be accessed by reviewers prior to publication using the Reviewer Key wveduewuxlknfmp.

Finally, our investigations of blastoderm explants indicate that PCP signaling functions downstream of Nodal in ML cell polarization and explant extension (Figure 5).

While we agree that it would be ideal to suppress the Nodal loss-of-function phenotype by activating PCP signaling, this experiment is not feasible to our knowledge. There are no reliable downstream targets of PCP signaling that can be reliably manipulated during gastrulation.

2) If CE is affected by loss of Nodal activity, a widening of the neural plate would be expected similar to the findings for the vangl2 MO. However, MZoep embryos actually have narrower neural plates than WT (Figure 2E, F), undermining the proposal that Nodal regulates CE.

This is an interesting point. The articles first describing the phenotypes of Nodal-deficient mutant zebrafish embryos (including MZ*oep*) note that extension is almost completely blocked, but that the other gastrulation movements of epiboly and convergence are intact (1, 2). As the reviewer points out, convergence of the neuroectoderm is apparent, but why this results in a narrower neural plate than WT is not known. Further, although convergence within MZ*oep* mutants doubtless occurs, previous reports demonstrate that the speed and persistence of converging neural plate cells within MZ*oep* mutants are significantly reduced compared to WTs at post-gastrulation stages (3), providing experimental evidence for disrupted convergence as well as extension. To better characterize convergence and extension movements in MZ*oep*-/- gastrulae and compare them quantitatively to WTs, we have re-analyzed our 4D nuclear tracking data to evaluate cell displacement in the mediolateral (convergence) and anteroposterior (extension) dimensions independently. By plotting these directional displacement data against a cell’s position within the embryo, characteristic slopes emerged representing convergence and extension movements, respectively. We also quantified absolute displacement along each axis in embryos of each genotype. Such analysis revealed that, although convergence was still apparent in MZ*oep*-/- mutants, cell displacement in the ML dimension was significantly reduced compared to WT. AP extension was also severely disrupted in terms of both absolute displacement and spatial organization of cell movements. These new analyses have been added to Figure 1 and are described within the Results section.

Also, why are the markers in Figure 2E more intensely stained in MZoep embryos compared to WT?

Our new RNA-seq data comparing WT and MZ*oep*-/- gastrulae demonstrate that *dlx3b* is expressed at slightly higher levels in MZ*oep* mutants, which may account for stronger staining. This is further compounded by increased cell density resulting from reduced extension of the neuroectoderm in these mutants (2). We have included these RNA-seq expression data in the new panel D in Figure 2—figure supplement 1, and discuss it briefly within the Results section.

3) The statement "Orientation of MZoep-/- cellular protrusions was also only partially improved within WT hosts (Figure 1I)" is not supported and should be adjusted, since the graphs in Figure 1H and I look similar. The finding that MZoep cells transplanted into wildtype hosts display nearly normal protrusion orientation is interesting. Does this depend on the polarization of the surrounding cells and would, for instance, wildtype ectoderm cells have a more random orientation of the protrusions when transplanted into MZoep?

To statistically analyze the orientation of protrusions, measured orientations were binned into one of 12 sectors (30° each, 360° total) which were then categorized as mediolateral (within 60° of horizontal), anteroposterior (within 60° of vertical), or neither (the remaining 120°). These bins have been color-coded on the rose diagrams of protrusive orientation in Figure 1M. The distribution of protrusions within each of these categories was then compared between experimental conditions using Chi-square tests. This analysis revealed that the distribution of protrusions in MZ*oep*-/- cells transplanted into WT hosts differed significantly from both MZ*oep*-/- to MZ*oep*-/- controls (p<0.0001) and WT to WT controls (p=0.0053), showing that while the polarity of their protrusions was improved by a WT environment, they did not align to the same degree as WT cells. These results are described within the Results section, and this analysis is further described in the Materials and methods section.

That MZ*oep*-/- cells transplanted into the neuroectoderm of WT hosts exhibit significantly improved alignment of their cell bodies and orientation of protrusions indeed demonstrates a non-autonomous effect on planar cell polarization. While it would be interesting to know whether this is likewise true in the reciprocal experiment (WT donors in MZ*oep*-/- hosts), we hesitated to do this experiment because the results would be too difficult to interpret. When WT cells are transplanted into MZ*oep* mutants they tend to become mesoderm, and it would be difficult to confirm that donor cells were neuroectodermal (so as to compare “oranges with oranges”).

4) Using the blastoderm explants, the authors suggest that convergence movements in this ex vivo system depend on Nodal signaling and that Nodal acts upstream of PCP signaling. However, the extension of the explant or the alignments of the cells aren't necessarily a readout of convergence movements. To claim this, it would be necessary to show actual convergence movements of the cells.

To address this concern, we have performed automated 4D cell tracking within nuclear labeled explants both with and without ectopic *ndr2*. These data were analyzed in the same fashion as tracking data from intact embryos, and yielded remarkably similar results. Cell tracks within *ndr2*-expressing explants were observed moving inward from lateral positions while moving outward along the axis of extension, tell-tale signs of C&E. Further, by graphing displacement along each axis against cell positions, we see slopes that are very similar to those in WT embryos, all highly suggestive of C&E as the primary morphogenetic mechanism of explant extension. These data have been included as new panels G-K in Figure 3 and described within the Results section.

Is it possible that elongation is promoted by a general increase in cell movements or differential proliferation?

To address the possibility of differential or localized proliferation, we manually annotated cell divisions within time-lapse series of nuclear labeled explants and plotted their locations with respect to the center of each explant. We found that cell divisions were concentrated along the axis of extension, and although this may partially reflect the elongated shape of *ndr2*-expressing explants, it suggests a possible role for cell divisions in ex vivo extension. However, the number of cell divisions during these stages of ex vivo development is relatively small. Furthermore, an article recently published in *eLife* demonstrated that blocking cell divisions in zebrafish blastoderm explants did not prevent their extension (4). We also did not observe a “hotspot” of proliferation that would indicate a region of outgrowth. Together these data suggest that differential proliferation may play a small role in explant extension, but is unlikely to contribute substantially. These data are shown in a new figure supplement, Figure 3—figure supplement 2, and described in the Results section.

Is Prickle-GFP localization polarized in explants?

To evaluate Pk-GFP localization in intact embryos, sparse labeling was achieved by transplanting cells from embryos injected with RNA encoding Pk-GFP (and H2B-RFP) into host embryos. Due to their small size, such transplantations between explants were not feasible. As an alternative approach, we attempted to achieve sparse labeling in explants through mosaic injections into one of sixteen blastomeres, but failed to observe reliable Pk-GFP puncta.

Furthermore, injection of Nodal ligands also induces the specification of endoderm and mesoderm, which are both lacking in the uninjected control. The activation of PCP signaling during the extension of the explant could be due to the presence of endoderm and mesoderm rather than due to the activity of Nodal signaling per se. Drug treatment to block Nodal signaling after acquisition of cell fate could be a possible experiment to discriminate whether continuous Nodal signaling activity is needed.

We in fact already performed this exact experiment, which was included in the second, co-submitted manuscript addressing mesoderm-independent roles for Nodal in C&E. Because this second manuscript was not recommended for publication, we moved these data into a new Figure 6 for the current revised manuscript. We indeed found that inhibition of Nodal signaling after mesoderm specification significantly reduces ex vivo extension, and further found that Nodal promotes extension of explants devoid of mesoderm marker expression. Together these results demonstrate that continuous Nodal signaling (per se) is required for full explant extension. This new Figure 6 is accompanied by 2 figure supplements, Figure 6—figure supplements 1 and 2, and is described in the Results section.

The paper would also benefit from a clearer distinction between elongation and CE. This is crucial since the effect of Nodal could in principle be mostly on prechordal plate migration rather than CE.

We agree that the distinction between extension and C&E is important, as morphogenetic mechanisms other than C&E certainly contribute to AP extension. However, we find it unlikely the prechordal plate (ppl) migration contributes to explant extension for the following reasons:

1) Highly extending explants from embryos injected with 10 pg of *ndr2* do not express markers for ppl, such as *gsc* (Figure 3—figure supplement 1).

2) We observed expression of *gsc* in explants from embryos injected with higher doses of *ndr2* (25-100pg), but this appears to be inversely correlated with their degree of extension (Figure 3—figure supplement 1), arguing against the notion that explant extension is driven by prechordal plate migration.

5) Nodal acts highly locally, not in a widespread manner throughout the embryo (see Figure 4—figure supplement 1 and Xu et al., 2014). How would Nodal regulate cell behavior far away in the neuroectoderm?

Van Boxtel at al., 2015 (5) demonstrate that there is low-level activation of downstream Nodal signaling within the zebrafish neural plate at gastrulation stages. Moreover, Sampath et al., 1998 (6), showed that *cyc/ndr2* is expressed in the axial mesoderm up to 90% epiboly – thus, just below the neuroectoderm undergoing C&E.

6) The conclusions rely on "significant differences" between experimental conditions determined by statistical tests. However, the effect sizes are often very small, and the variance within experimental groups is high. The low p values are a consequence of the very high sample numbers, but the biological significance of the determined effect sizes remains unclear. The authors should therefore apply bootstrapping or resampling methods using subsets from the data distributions to test whether the effect sizes are reproducible.

As in any complex living system, cell shape, alignment, and movements are (expectedly) highly variable within live embryos. This is why we have quantified – as the reviewer noted – a large number of cells and/or embryos for every experiment. However, we disagree that the resulting effect sizes are small or insignificant. Although the difference between an orientation of 20° and 45° may not sound substantial, it is the difference between a highly polarized (20°) and randomly oriented (45°) cell. And while not every cell in even a WT embryo will achieve this level of polarization, our statistics demonstrate that *as a population,* these cells are highly polarized and mutant cells are significantly less so. In the relevant figures, we included individual data points for all of our cell and explant measurements to be transparent about exactly how variable this type of data is. However, these plots can be misleading, and suggest, for example, that populations of cells are less well polarized than they are. To better represent differences between groups, in the revised manuscript we have changed our graphical representations of cell shape and alignment (in Figures 1, 2, and 5) to violin plots. Individual data points are still included, but we feel these plots more clearly illustrate the “shape” of each dataset as a whole and highlight important differences between them. One example is shown in Figure 1J.

7) It is unclear why embryos uniformly injected with ndr2 give rise to polarized explants. It would be good to show that the presence of asymmetry does not depend on asymmetric injection of Nodal ligands, e.g. by executing in situ hybridization stainings with probes for injected ndr2 (perhaps with a GFP tag or the like to distinguish it from endogenously produced Nodals).

We indeed speculate that small asymmetries resulting from injection are amplified by feed-back and feed-forward Nodal signaling (7), as it is not possible to ensure perfectly uniform distribution of injected RNA. This is because 1) we are injecting a small 1 nL volume into an embryo with a 600 mm diameter, and we will rarely hit the exact center, and 2) we have no control over cytoplasmic streaming, which distributes cytoplasm and injected RNAs between the cells of the early embryo. We have included a small section in the Discussion section describing this speculated source of asymmetry (fifth paragraph). While we are certainly curious to discover the origins of this asymmetry in future studies, we do not feel that how the asymmetry arises in any way affects our interpretation of the data presented nor any conclusions drawn therefrom.

If injected ndr2 is uniformly distributed, is Nodal autoactivation required for polarization, i.e. what happens in MZsqt;cyc embryos?

Although we speculate that in our injected experimental embryos Nodal may not be evenly distributed, we would be similarly excited to perform this experiment to dissect the role of Nodal autoactivation. However, given that this experiment would require the arduous task of germline transplantation to make MZ mutants, we had neither the time nor resources to complete it in a timely manner.

8) The authors state that "Although the PCP genes wnt5b, vangl2 (trilobite), and gpc4 (knypek) were expressed at apparently normal levels in MZoep-/- gastrulae by whole mount in situ hybridization (WISH) (Figure 2—figure supplement 1B)", but the data does not show this; instead all genes seem to be upregulated in MZoep embryos. Please adjust the conclusions.

As mentioned above, we have since included RNA-seq data demonstrating that most PCP genes, including *wnt5b,vangl2*, and *gpc4*, are expressed at normal levels in MZ*oep* mutants.

References:

1) Feldman B, Gates MA, Egan ES, Dougan ST, Rennebeck G, Sirotkin HI, et al. Zebrafish organizer development and germ-layer formation require nodal-related signals. Nature. 1998;395(6698):181-5.

2) Gritsman K, Zhang J, Cheng S, Heckscher E, Talbot WS, Schier AF. The EGF-CFC protein one-eyed pinhead is essential for nodal signaling. Cell. 1999;97(1):121-32.

3) Araya C, Tawk M, Girdler GC, Costa M, Carmona-Fontaine C, Clarke JD. Mesoderm is required for coordinated cell movements within zebrafish neural plate in vivo. Neural Dev. 2014;9:9.

4) Schauer A, Pinheiro D, Hauschild R, Heisenberg C-P. embryonic explants undergo genetically encoded self-assembly. *eLife*. 2020e55190.

5) van Boxtel AL, Chesebro JE, Heliot C, Ramel MC, Stone RK, Hill CS. A Temporal Window for Signal Activation Dictates the Dimensions of a Nodal Signaling Domain. Dev Cell. 2015;35(2):175-85.

6) Sampath K, Rubinstein AL, Cheng AM, Liang JO, Fekany K, Solnica-Krezel L, et al. Induction of the zebrafish ventral brain and floorplate requires cyclops/nodal signalling. Nature. 1998;395(6698):185-9.

7) Müller P, Rogers KW, Jordan BM, Lee JS, Robson D, Ramanathan S, et al. Differential diffusivity of Nodal and Lefty underlies a reaction-diffusion patterning system. Science. 2012;336(6082):721-4.